Manuscript prepared for Atmos. Chem. Phys.
with version 2015/04/24 7.83 Copernicus papers of the LATEX class copernicus.cls.
Date: 26 January 2018

# Contributions of natural and anthropogenic sources to ambient ammonia in the Athabasca Oil Sands and north-western Canada

Cynthia Whaley[1], Paul A. Makar[1], Mark W. Shephard[1], Leiming Zhang[1], Junhua Zhang[1], Qiong Zheng[1], Ayodeji Akingunola[1], Gregory R. Wentworth[2,3], Jennifer G. Murphy[2], Shailesh K. Kharol[1], and Karen E. Cady-Pereira[4]

[1]Air Quality Research Division, Environment and Climate Change Canada, 4905 Dufferin Street, Toronto, Ontario, Canada
[2]Dept of Chemistry, University of Toronto, 80 St George Street, Toronto, Ontario, Canada
[3]Environmental Monitoring and Science Division, Alberta Environment and Parks, 9888 Jasper Ave NW, Edmonton, Alberta, Canada
[4]Atmospheric and Environmental Research (AER), Lexington, Massachusetts, USA

*Correspondence to:* Cynthia Whaley (cynthia.whaley@canada.ca)

**Abstract.** Atmospheric ammonia ($NH_3$) is a short-lived pollutant that plays an important role in aerosol chemistry and nitrogen deposition. Dominant $NH_3$ emissions are from agriculture and forest fires, both of which are increasing globally. Even remote regions with relatively low ambient $NH_3$ concentrations, such as northern Alberta and Saskatchewan in northern Canada, may be of interest

because of industrial Oil Sands emissions and a sensitive ecological system. A previous attempt to model $NH_3$ in the region showed a substantial negative bias compared to satellite and aircraft observations. Known missing sources of $NH_3$ in the model were re-emission of $NH_3$ from plants and soils (bidirectional flux) and forest fire emissions, but the relative impact of these sources on $NH_3$ concentrations was unknown. Here we have used a research version of the high-resolution air quality

forecasting model, GEM-MACH, to quantify the relative impacts of semi-natural (bidirectional flux of $NH_3$ and forest fire emissions) and direct anthropogenic (Oil Sands operations, combustion of fossil fuels, and agriculture) sources on ammonia volume mixing ratios, both at the surface and aloft, with a focus on the Athabasca Oil Sands region during a measurement-intensive campaign in the summer of 2013. The addition of fires and bidirectional flux to GEM-MACH has improved the model

bias, slope and correlation coefficients relative to ground, aircraft, and satellite $NH_3$ measurements significantly.

By running the GEM-MACH-Bidi model in three configurations and calculating their differences, we find that averaged over Alberta and Saskatchewan during this time period an average of 23.1% of surface $NH_3$ came from direct anthropogenic sources, 56.6% (or 1.24 ppbv) from bidirectional

flux (re-emission from plants and soils), and 20.3% (or 0.42 ppbv) from forest fires. In the $NH_3$ total column, an average of 19.5% came from direct anthropogenic sources, 50.0% from bidirectional flux, and 30.5% from forest fires. The addition of bidirectional flux and fire emissions caused the

overall average net deposition of $NH_x$ across the domain to be increased by 24.5%. Note that forest fires are very episodic and their contributions will vary significantly for different time periods and regions.

This study is the first use of the bidirectional flux scheme in GEM-MACH, which could be generalized for other volatile or semi-volatiles species. It is also the first time CrIS satellite observations of $NH_3$ have been used for model evaluation, and the first use of fire emissions in GEM-MACH at 2.5-km resolution.

## 1 Introduction

Ammonia ($NH_3$) is a short-lived pollutant that is receiving global attention because of its increasing concentrations. Emissions of $NH_3$ – which are in large part from agricultural fertilizer, livestock (Behera et al., 2013; Environment and Climate Change Canada, 2016), and biomass burning (Olivier et al., 1998; Krupa, 2003) – have not been regulated to the same extent as other nitrogen species. $NH_3$ is the only aerosol precursor whose global emissions are projected to rise throughout the next century (Moss et al., 2010; Lamarque et al., 2010; Ciais et al., 2013).

$NH_3$ has an atmospheric lifetime of hours to a day (Seinfeld and Pandis, 1998; Aneja et al., 2001). It is a base that reacts in the atmosphere with sulphuric acid ($H_2SO_4$) and nitric acid ($HNO_3$) to form crystalline sulphate, nitrate salts (e.g., $(NH_4)_2SO_4$, $NH_4HSO_4$, $NH_4NO_3$) and aqueous ions ($SO_4^{2-}$, $HSO_4^-$, $NO_3^-$), (Nenes et al., 1998; Makar et al., 2003) which are significant components of fine particulate matter ($PM_{2.5}$) (e.g., Jimenez et al., 2009, Environment Canada, 2001), thus having health (Pope III et al., 2002; Lee et al., 2015) and climate impacts (IPCC, 2013). A large portion of $NH_3$ is readily deposited in the first 4-5 km from its source, but when in fine particulate form (as $NH_4^+$), its lifetime is days to several weeks (Galperin and Sofiev, 1998; Park et al., 2004; Behera et al., 2013; Paulot et al., 2014), and it can be transported hundreds of kilometers (Krupa, 2003; Galloway et al., 2008; Makar et al., 2009). Deposition of $NH_3$ and these aerosols can lead to nitrogen eutrophication and soil acidification (Fangmeier et al., 1994; Sutton et al., 1998; Dragosits et al., 2002; Carfrae et al., 2004). $NH_3$ is listed as a Criteria Air Contaminant (Environment and Climate Change Canada, 2017) in order to help address air quality issues such as smog and acid rain.

Modelling can be used to better understand $NH_3$ processes. Recent $NH_3$ models have focused on improving bidirectional flux processes and impacts of livestock. Measurements of $NH_3$ bidirectional flux include those in Farquhar et al. (1980); Sutton et al. (1993, 1995); Asman et al. (1998); Nemitz et al. (2001), with indirect support for bidirectional flux also in Ellis et al. (2011). Thus, these studies were the motivation for the recent design of parameterizations to describe this important process (Wu et al., 2009; Wichink Kruit et al., 2010; Massad et al., 2010; Zhang et al., 2010; Zhu et al., 2015; Fu et al., 2015; Hansen et al., 2017). Additionally, satellite observations are providing valuable insight on ammonia concentrations and emissions both on regional and global scales

(Beer et al., 2008; Clarisse et al., 2009; Shephard et al., 2011; Shephard and Cady-Pereira, 2015; Van Damme et al., 2014; Zhu et al., 2013).

The Athabasca Oil Sands region (AOSR), located in the north-eastern part of the province of Alberta, Canada, is a large source of pollution to air (Gordon et al., 2015; Liggio et al., 2016; Li et al., 2017) and ecosystems (Kelly et al., 2009; Kirk et al., 2014; Hsu et al., 2016), as well as a source of greenhouse gases (Charpentier et al., 2009) due to mining and processing by the oil industry. While $NH_3$ volume mixing ratios (VMRs) surrounding the AOSR in northern Alberta and Saskatchewan re-

main relatively low – around 0.6-1.2 ppbv background (this study and Shephard et al., 2015) – due to low population and lack of agriculture, the northern Alberta and Saskatchewan ecosystems are sensitive to nitrogen deposition (Clair and Percy, 2015; Wieder et al., 2016a, b; Vitt, 2016; Makar et al., 2017), and the modelled background $NH_3$ must be correct in order to understand the relative impacts of the oil sands operations. It is important to understand if the AOSR facilities or other sources

(e.g., fires, re-emissions) are causing $NH_3$ to reach levels that cause ecosystem damage. A monitoring study from 2005 to 2008 found $NH_3$ VMRs near Fort McMurray and Fort McKay (population centers in the vicinity of the oil sands facilities) to be highly variable in space and time with a range of 1.1 to 8.8 ppbv (where the upper end corresponds to $NH_3$ levels found in agricultural regions of Canada and the U.S.), with $NH_3$ concentrations 1.5-3× higher than $HNO_3$ concentrations

(Bytnerowicz et al., 2010). Hsu and Clair (2015) also found $NH_3$ concentrations in the AOSR to be much higher than $HNO_3$, $NO_3^-$, and $NH_4^+$ concentrations (by  5, 23, and 1.8×, respectively). Thus, $NH_3$ may contribute the largest fraction of deposited nitrogen in the AOSR compared to other nitrogen species. Estimates of deposition of nitrogen compounds in the AOSR are described in Makar et al. (2017), however they did not include $NH_3$ bidirectional flux or forest fires in their

model simulations.

In a previous study by Shephard et al. (2015) it was found that the GEM-MACH air quality forecasting model (Moran et al., 2010, 2013; Makar et al., 2015a, b; Gong et al., 2015), using a domain covering the Canadian provinces of Alberta and Saskatchewan, at 2.5-km resolution, under-predicted summertime tropospheric ammonia VMRs by 0.4-0.6 ppbv (which is 36-100 % depending on alti-

tude - see Fig. 16 in Shephard et al., 2015) in the AOSR when compared to Tropospheric Emission Spectrometer (TES) satellite measurements and aircraft measurements. Having too much modelled $NH_x$ deposition is a cause that was ruled out when Makar et al. (2017) showed that GEM-MACH actually underestimates $NH_x$ deposition. Underestimating anthropogenic and agricultural emissions was also ruled out as a cause since the GEM-MACH model performs well in southern Canada and

the U.S when compared to the U.S. Ambient Ammonia Monitoring Network (AMoN). $NH_3$ sources known to be missing from the GEM-MACH model were forest fire emissions and re-emission of deposited $NH_3$ from soils and plants (the latter referred to as bidirectional flux, hereafter), which would have the greatest impact in background areas, such as northern Alberta and Saskatchewan. Therefore, these two sources were added to an updated version of GEM-MACH and model simu-

lations were repeated for a 2013 summer period (12 August to 7 September 2013) during which an intensive measurement campaign occurred. We utilize ground, aircraft and satellite measurements of $NH_3$ and related species to evaluate the model and to quantify the impacts of the different sources on atmospheric $NH_3$ and its deposition.

  Section 2 provides the model description. Section 3 provides a brief description of ammonia mea-
surements during the campaign. Section 4 presents the evaluation of three model scenarios against three different types of measurements (surface, aircraft, and satellite), and Section 5 presents our quantitative assessment on the impacts of different sources of $NH_3$ to ambient VMRs and $NH_x$ fluxes in the region. Our conclusions appear in Section 6.

## 2 GEM-MACH model description

GEM-MACH (Global Environment Multiscale-Modelling Air quality and CHemistry) is an on-line chemical transport model, which is embedded in GEM, Environment and Climate Change Canada (ECCC)'s numerical weather prediction model (Moran et al., 2010). This means that the chemical processes of the model (gas-phase chemistry, plume rise emissions distribution, vertical diffusion and surface fluxes of tracers, and a particle chemistry package including particle microphysics, cloud
processes, and inorganic heterogeneous chemistry) are imbedded within GEM's physics package, which in turn is imbedded within GEM's dynamics package, the latter handling chemical tracer advection. A detailed description of the process representation of GEM-MACH, and an evaluation of its performance for pollutants such as ozone and particulate matter (PM) appears in Moran et al. (2013); Makar et al. (2015a, b); and Gong et al. (2015).

GEM-MACHv2 is used operationally to issue twice-daily, 48-hour public forecasts of criteria air pollutants (ozone, nitrogen oxides, PM), as well as the the Air Quality Health Index (https://ec.gc.ca/cas-aqhi/). Any improvements to $NH_3$ in the model may result in better AQHI predictions, since $NH_3$ is a major precursor of $PM_{2.5}$, as mentioned in the introduction. We start with a similar, research version of GEM-MACHv2 (rev2285) to make the bidirectional flux modifications. The key differences be-
tween this and older versions are the use of a more recent meteorological package (GEMv4.8), the capability to nest in the vertical dimension as well as the horizontal dimension, and improvements to the treatment of fluxes, vertical diffusion, and advection.

  GEM-MACH can be run for many different spatial domains, at various spatial resolutions, and in 2-bin or 12-bin aerosol size distribution modes. For this study we run the model in the 2-bin mode
(for computational efficiency), using a nested set of domains. The outer domain at 10-km resolution covers North America, and the inner domain at 2.5-km resolution covers the provinces of Alberta and Saskatchewan. The latter is referred to as the 2.5-km Oil Sands domain. This set up, along with the emissions described in the next section is hereafter called our "base" simulation.

### 2.1 Emissions

The emissions of 25 species ($SO_2$, $SO_4$ (gas), sulphate, nitrate, $NH_4^+$, NO, $NO_2$, $NH_3$, CO, nitrous acid, benzene, propane, higher alkanes, higher alkenes, ethene, toluene, aromatics, formaldehyde, aldehydes, methyl ethyl ketone, creosol, isoprene, crustal material, elemental carbon, and primary carbon) used in GEM-MACH (base case) come from Canadian and U.S. emissions inventories: 2011 National Emissions Inventory (NEI) version 1 for U.S. emissions, and the Air

Pollutant Emission Inventory (APEI) 2013 for Canadian emissions (2010 for onroad and offroad emissions). Emissions were processed with SMOKE (Sparse Matrix Operator Kernel Emissions, https://www.cmascenter.org/smoke/) to convert the inventories into model-ready gridded hourly emissions files for modeling, separated into major point emissions (typically industrial emissions from stacks, emitted into the model layers that correspond to the stack height, at the reported temperature

and velocity in the inventory's stack parameters), and area emissions (emissions from spread-out sources, such as transportation and agriculture, emitted into the first model layer). For more details about these emissions, see Moran et al. (2015) and Zhang et al. (2017).

The emissions data for $NH_3$ from oil sands sources are reported to the Canadian National Pollutant Release Inventory (NPRI) on a "total annual emissions per facility" basis. $NH_3$ emissions are gen-

erally more uncertain than $SO_2$ and $NO_x$ emissions because $NH_3$ emissions are not measured to the same extent as those two. The oil sands represent only $1\%$ of total Alberta $NH_3$ emissions, at approximately 1438 tonnes in 2013. For comparison, about $18\times$ more $NO_x$ and $57\times$ more $SO_2$ was emitted from the oil sands facilities that year (http://www.ec.gc.ca/inrp-npri/donnees-data/index.cfm?lang=En). However, we found an issue with $NH_3$ in this inventory that impacted our model evaluation in the

region, which we describe below.

If stack parameters (e.g., stack height and diameter, volume flow rates, temperatures, etc.) are included as part of those NPRI data, then the emissions are allocated to large stacks in our configuration of the SMOKE emissions processing system. In the absence of this information, SMOKE will assign default stack parameters based on its source category code. For the Syncrude Canada Ltd. -

Mildred Lake Plant Site, NPRI ID 2274 (a facility in the AOSR), the default stack parameters were: 18.90 m for the stack height (which is within the first model layer), 0.24 m for the stack diameter, 320.0 K for the exhaust temperature, and 0.58 m/s for the exhaust velocity. However, when these defaults were applied to $NH_3$ emissions in initial model simulations, they were found to result in erroneous short term plume events with simulated surface $NH_3$ levels up to 2 orders of magnitude

higher than ground observations, and modelled VMRs aloft too low compared to aircraft measurements. Conversely, for species such as $SO_2$, for which stack parameters were reported, the model was able to correctly place the $SO_2$ enhancements in space and time, relative to observations. When we applied those same stack parameters for $NH_3$ emissions as well (stack height=183 m, stack diameter=7.9 m, exit temperature=513 K, exit velocity=23.9 m/s, from the NPRI website), the simulation

of surface $NH_3$ was greatly improved. All subsequent simulations reported here make use of this

correction, and we advise the reporting of stack parameters for all species for future inventories, in order to avoid this kind of error for models.

### 2.2 Ammonia bidirectional flux parameterization

$NH_3$ can be both deposited from the atmosphere to the ground, and re-emitted from soils and plants
back to the atmosphere. The two taken together are called bidirectional flux, since the flux of $NH_3$ can go both up and down. The source of $NH_3$ available for re-emissions are from the accumulated $NH_x$ in the soil and stomatal water, which can arise from increased deposition from anthropogenic sources, as well as from organic nitrogen decomposition (Booth et al., 2005), $N_2$-fixation (Vile et al., 2014), and natural microbial action (McCalley and Sparks, 2008).

The bidirectional flux scheme of Zhang et al. (2010) was applied within the GEM-MACHv2 model, replacing the original deposition velocity for $NH_3$ only (deposition velocity of other gas species follows a scheme based on a multiple resistance approach and a single-layer "big leaf" approach (Wesely, 1989; Zhang et al., 2002; Robichaud and Lin, 1991; Robichaud, 1994)). The bidirectional flux scheme is described in detail in Zhang et al. (2010), but we summarize it here.

Bidirectional exchange occurs between air-soil and air-stomata interfaces. The bidirectional flux $(F_t)$ equation is:

$$F_t = -\frac{C_a - C_c}{R_a + R_b} \tag{1}$$

where $R_a$ and $R_b$ are the aerodynamic and quasi-laminar resistances, respectively. $C_a$ is the $NH_3$ concentration in the air, and $C_c$ is the canopy compensation point concentration, given by Eq. (2).

$$C_c = \frac{\frac{C_a}{R_a+R_b} + \frac{C_{st}}{R_{st}} + \frac{C_g}{R_{ac}+R_g}}{(R_a + R_b)^{-1} + (R_{st})^{-1} + (R_{ac} + R_g)^{-1} + (R_{cut})^{-1}} \tag{2}$$

where $C_{st}$ and $C_g$ are the stomatal and ground compensation points, and $R_i$ are the resistances in s/m of the ground/soil ($R_g$), stomata ($R_{st}$), cuticle ($R_{cut}$), and in-canopy aerodynamic ($R_{ac}$). All resistance formulas can be found in Zhang et al. (2003).

Stomata (st) and ground (g) compensation points are both calculated using Eq. (3):

$$C_{st,g} = \frac{A}{T_{st,g}} \exp\left(\frac{-B}{T_{st,g}}\right)\Gamma_{st,g} \tag{3}$$

$A$ and $B$ are constants derived from the equilibria constants for $NH_3(g)$ in leaves' stomatal cavities to $NH_4^+$ and $OH^-$ in the water contained in the apoplast within the leaf and in the soil where $NH_3(g)$ in the soil pore air space is in equilibrium with the $NH_4^+$ and $OH^-$ dissolved in soil water (Pleim et al., 2013). A=161500 mol K/L (Nemitz et al., 2000), or $2.7457 \times 10^{15}$ ugK/m$^3$ (Pleim et al., 2013) for
$NH_3$ for both stomata and soil. B=10380 (Nemitz et al., 2000). $\Gamma_{st,g}$ is the emission potential of the stomata and ground, respectively and, in theory, is equal to the $NH_4^+$ concentration over the $H^+$ concentration in the apoplast water of the canopy leaves or soil water:

$$\Gamma_{st,g} = \frac{[NH_4^+]_{st,g}}{[H^+]_{st,g}} \tag{4}$$

However, since there are no modeled $NH_4^+$ and $H^+$ apoplast water concentrations to use, we use $\Gamma_{st,g}$ from Wen et al. (2014), which is based on long-term empirical averages. Wen et al. (2014) gives a range of values for emission potentials for 26 land use categories (LUCs), and we use the low-end of the values in our model with the following exceptions: We further lower the $\Gamma_g$ for agriculture LUCs to 800, and increase $\Gamma_{st}$ of boreal forest LUCs to 3000, all of which were necessary in order to achieve realistic $NH_3$ concentrations (e.g., compared to reported AMoN values), while staying consistent with $\Gamma$ findings from the literature.

This version of the model, which we call GEM-MACH-Bidi (or just "bidi" hereafter) was quite sensitive to the selection of these emission potentials, which are themselves highly uncertain (Wen et al., 2014). GEM-MACH-Bidi uses the exact same emissions as in the base case, described in the previous section. However, when the sign of $F_t$ in Eq. (1) becomes positive (that is, when $C_a < C_c$), the bidirectional flux acts effectively as an additional source of $NH_3$ gas, releasing stored $NH_3$ until and unless the ambient concentration rises to the compensation point concentration. When the flux is negative, net deposition of $NH_3$ occurs.

It is important to note that $C_{st,g}$ values are exponentially dependent on temperature (Fig. 1 shows an example of this relationship for the dominant LUCs in the northern part of the domain), and the higher the compensation point, the greater the likelihood there will be upward flux. The lower the $C_{st,g}$, the more likely there will be deposition. Since our simulation period was August and September 2013, when the average temperature in the AOSR was about $18°C$ (agriculture.alberta.ca/acis/alberta-weather-data-viewer.jsp), we expect to have more $NH_3$ re-emission than at other times of the year. During the rest of the year (e.g., the preceeding winter and spring), the compensation point would be much lower, greatly increasing the likelihood to have net deposition, even in northern Alberta/Saskatchewan where ambient $NH_3$ concentrations are low. Other meteorological factors affect the magnitude of bidirectional flux via the resistance terms. For example canopy compensation points have been observed to decrease with decreasing wind velocity, and increased precipitation (Flechard and Fowler, 1998; Fowler et al., 1998; Biswas et al., 2005; Zhang et al., 2010). In other words, we expect more re-emission during higher winds and drier conditions.

Other chemical transport models, such as GEOS-Chem and CMAQ use a similar method as Zhang et al. (2010), however, instead of the constant average soil emission potentials used here, they utilize a CMAQ-agroecosystem coupled simulation to calculate a soil pool from which to estimate $\Gamma_g$ (Bash et al., 2013; Pleim et al., 2013; Zhu et al., 2015). In this case, the emission potential will vary and can go to zero if the $NH_4^+$ in the pool is depleted. However, it was shown in Wen et al. (2014) that their $\Gamma_{st,g}$ worked well during the same time of year as this investigation (August and September). This time of year was also shown in Zhu et al. (2015) to not have a large effect on emissions from the $NH_4^+$ pool. Additionally, Wentworth et al. (2014) calculated the approximate relative abundances of $NH_x$ in the boundary layer versus $NH_4^+$ in the soil pool to assess whether surface-to-air fluxes were sustainable. They found that soil $NH_4^+$ concentrations were much greater than

boundary layer $NH_x$ (by over two orders of magnitude), further supporting the assumption made here. In addition, the turnover time for soil $NH_4^+$ is on the order of one day, hence it is unlikely that $NH_3$ bi-directional fluxes would significantly deplete/enhance soil $NH_4^+$ pools. Finally, given that GEM-MACH is used for real-time air quality forecasts at ECCC, it is not desirable for our bidirectional flux scheme to have to rely in advance on another model's output. Therefore, we use this simplified version, and assess whether its results provide an improvement (smaller biases and better correlations to measurements) to simulated $NH_3$.

### 2.3  Addition of forest fire emissions

Our third model scenario (called "fire+bidi" hereafter) uses the GEM-MACH-Bidi model, and the exact same area emissions and anthropogenic major point emissions as the base and bidi scenarios. However, in addition, we add hourly North American forest fire emissions for all species to the major point emissions. The forest fire emissions system for GEM-MACH (called "Firework") is described in detail in Pavlovic et al. (2016). Briefly, to calculate the fire emissions for input to FireWork, biomass burning areas are first identified in near real time by the Canadian Wildland Fire Information System (CWFIS), which is operated by the Canadian Forest Service (http://cwfis.cfs.nrcan.gc.ca/home). CWFIS uses fire hotspots detected by NASA's Moderate Resolution Imaging Spectroradiometer (MODIS) and NOAA's Advanced Very High Resolution Radiometer and Visible Infrared Imaging Radiometer Suite imagery as inputs. Daily total emissions per hotspot are then estimated by the Fire Emission Production Simulator module of the BlueSky Modeling Framework (Larkin et al., 2009). SMOKE was then used to prepare model-ready hourly emissions of several species (including $NH_3$) in a point-source format for model input.

In ECCC's operational forest fire forecasts, these emissions are used at 10-km resolution for the domain encompassing North America, with forest fires being treated as point sources with specific plume rise (Pavlovic et al., 2016). We have added 2013 forest fire emissions which were originally created for the 2013 Firework forecasts to the anthropogenic point source emissions used in the base case simulation, and have modified the GEM-MACH model to be able to accommodate the changing number of major point sources each day (as the number of fires changes daily). Fire plume rise is an ongoing area of investigation (e.g., Heilman et al., 2014; Paugam et al., 2016); smoldering emissions tend to be emitted directly at the surface, whereas flaming emissions can inject plumes to the upper troposphere. Here, we have set all fire emissions to be distributed evenly throughout the boundary layer, which is a simplification, but one that averages out smouldering and flaming plume heights. Different parameterizations of fire plume rise are currently under development in GEM-MACH. The Fireworks fire emissions are described in detail in Zhang et al. (2017), and this study represents the first time they have been used at a 2.5-km horizontal resolution.

## 2.4 Model setup for three scenarios

The base, bidi, and fire+bidi models were all run in the following way: Each scenario was run from 1 August to 7 September, 2013, where the first 11 days were "spin up" in order to allow chemical concentrations to stabilize, and are not used in our evaluation. This is a sufficient amount of spinup time, given that the atmospheric lifetime of $NH_3$ is typically up to 1 day (Seinfeld and Pandis, 1998; Aneja et al., 2001), and given that it is close to the transport time of air crossing the larger North American domain. The time period from 12 August to 7 September was chosen to coincide with the intensive measurement campaign described in Section 3.

The model was run in a nested setup, whereby the North American domain was run at 10-km resolution using "climatological" chemical initial and boundary conditions from a 1-year MOZART simulation for all pollutants (Giordano et al., 2015). The nested Oil Sands region (which covers most of Alberta and Saskatchewan) was run at 2.5-km horizontal resolution, using the initial and boundary conditions from the 10-km North American model run. Figure 2 shows the two model domains.

The model simulations for the pilot and nested domains were not run as a continuous multiday forecast, but rather following the operational air quality forecast process, where the meteorological values are updated regularly with new analyses (products of meteorological data assimilation, which provide optimized initial conditions for the 12 UTC hour of each day. The analyses were obtained from ECCC archives (Buehner et al., 2013, 2015; Caron et al., 2015)), in order to prevent chaotic drift of the model meteorology from observations. Consequently, our simulation setup comprises simulations on the North American domain in 30-hour cycles starting at 12 UTC, and the Oil Sands domain in 24-hour cycles starting at 18 UTC (the 6 hour lag being required to allow meteorological spinup of the lower resolution model). The next cycle uses the chemical mass mixing ratios from the end of the last cycle as initial conditions for the next 24-30 hours. This system of staggered meteorological driving forecasts with a continuous chemical record continues until the full time period completes.

We run GEM-MACH in the 2-bin particle mode, which means that particles fall in either fine mode (diameter 0-2.5 $\mu$m) or coarse mode (diameter 2.5-10 $\mu$m), for computational efficiency (although sub-binning is used in some particle microphysics processes in order to ensure an accurate representation of particle microphysics (Moran et al., 2010)), and in order to follow the setup used for the operational 10-km resolution GEM-MACH forecast.

## 3 Measurements

Our three model simulations (base, bidi, and fire+bidi) are evaluated with surface, aircraft, and Cross-track Infrared Sounder (CrIS) satellite measurements. We briefly describe each of these observation datasets below.

### 3.1 AMS13 ground measurements

An extensive suite of instrumentation was deployed at monitoring site AMS13 (57.1492°N, 111.6422°W, 270 m.a.s.l., Fig. 3) from 7 August 2013 until 12 September 2013. Mining operations and bitumen upgrading facilities are 5 km to the south and north of the site, which is surrounded by boreal forest, with dominant winds from the west, averaging 1.9 m/s throughout the year. The average temperatures in the region for August are highs in the low 20s°C, and lows around 10°C, which is warm enough to make upward $NH_3$ flux more likely (recall Fig. 1), but temperatures drop rapidly at the end of August, into September, where the September highs average around 15°C, lows around 5°C. The skies are the clearest during August, with at least partly clear skies 50% of the time. That said, the warm season (May through September) is the wetter season (average of 20% chance of precipitation daily), with more precipitation than during the cold season (when there is an average of 7% chance of precipitation daily), but year-round precipitation, as well as relative humidity, are both relatively low in the AOSR. During the cold season (November through February), the average temperatures range from -21°C to -5°C, when the forest and soils are more likely to be a deposition sink for $NH_3$. During November to April, it is also much cloudier, with February having cloudy conditions 77% of the time. (All weather data cited here are from the annual report at Fort McMurray, found here: https://weatherspark.com/y/2795/Average-Weather-in-Fort-McMurray-Canada-Year-Round).

$NH_3$, fine particulate ammonium and nitrate, and other species were measured by the Ambient Ion Monitor-Ion Chromatograph (AIM-IC), via an inlet 4.55 m off the ground. The uncertainty of these measurements is $\pm$ 15%. These measurements are described in more detail in Markovic et al. (2012).

Data gaps sometimes appeared in the surface $NH_3$ time series for the following reasons: instrument zero (Aug 14/15 and 17/18), instrument maintenance (Aug 19) and a power outage (Aug 27/28).

### 3.2 Aircraft measurements

During the Oil Sands Monitoring Intensive campaign, there were a total of 22 flights spanning 13 August to 7 September 2013. These measurements are described in detail in Shephard et al. (2015); Gordon et al. (2015); Liggio et al. (2016); Li et al. (2017), and are summarized here. Aircraft $NH_3$ measurements were conducted with a dual quantum cascade laser (QCL) trace gas monitor (Aerodyne Inc., Billerica, MA, USA; McManus et al., 2008), collecting data every 1 s. Outside air was sampled through a heated Teflon inlet tube shared with a high-resolution time-of-flight chemical ionization mass spectrometer (HR-ToF-CIMS); the flow rate through the QCL was 10.8 L min$^{-1}$. The 1 $\sigma$ uncertainty for each measurement was estimated to be $\pm$0.3 ppbv ( $\pm$35%) (Shephard et al., 2015).

Particulate $NH_4^+$ (0- <1 $\mu$m in diameter) was measured by the Aerodyne high-resolution time-
of-flight aerosol mass spectrometer (HR-ToF-AMS) instrument on board the same flights, which
collected data every 10 s. The ambient air was drawn through a forward facing, shrouded isoki-
netic particle inlet from which the HR-ToF-AMS sub-sampled. The total residence time in the in-
let and associated tubing was approximately 1 second. The error on these measurements is $\pm 9\%$.
(Liggio et al., 2016)

Figure 3 shows a sample flight path from the campaign from 13 August 2013 – one of the thirteen
flights with valid $NH_3$ measurements. The others took place on 15-17, 19 (two this day), 22-24,
26, 28 August, and 5-6 September 2013. $NH_3$ data on the other nine flights were invalidated due to
instrument issues (those on 14, 20-21, 29, 31 August, and 2-4 September 2013), but were successful
for the $NH_4^+$ measurements.

### 3.3 CrIS satellite measurements

CrIS was launched in late October 2011 on board the Suomi NPP platform. CrIS follows a sun-
synchronous orbit with a daytime overpass time at 13:30 (ascending) and a night time equator over-
pass at 1:30 (descending), local time. The instrument scans along a 2200 km swath using a 3 x 3
array of circular pixels with a diameter of 14 km at nadir for each pixel. The CrIS Fast Physical Re-
trieval (CFPR) described by Shephard and Cady-Pereira (2015) is used to perform satellite profile
retrievals of $NH_3$ VMR given the infrared emission spectrum from the atmosphere. This retrieval
uses an optimal estimation approach (Rogers, 2000) that provides the satellite vertical sensitivity
(averaging kernels) and an estimate of the total errors (error covariance matrix).

We take the CrIS retrieved profile and match it up with the closest model profile in both distance
and time, compute the distance between the CrIS pixel and model field for each time step, and then
select the time step that best matches the satellite overpass time. Since the model time steps are
every hour with a 10-km spatial resolution they are always matched up to better than half an hour,
and within 5 km.

### 4 Model evaluation

An older version of GEM-MACH (v1.5.1) has been compared to TES satellite and aircraft mea-
365 surements of ammonia over the AOSR (Shephard et al., 2015). Simulations with that version of the
model were shown to be biased low, by about -0.5 ppbv, throughout the lower-tropospheric verti-
cal profile. This represented a substantial deficit in the model predicted sources of $NH_3$, prompting
the current work. We now compare our three GEM-MACHv2 simulations (base, bidi, and fire+bidi)
against surface point measurements at the measurement site near an oil sands facility (AMS13),
aircraft measurements over the broader AOSR, and satellite measurements over the Alberta and

Saskatchewan area. We will discuss which simulation agrees best with measurements and where there may still be room for additional model improvement.

## 4.1 At the AMS13 ground site

Figure 4 shows the timeseries of the daily average (for clarity) VMRs of $NH_3$ and concentrations of fine-particulate $NH_4^+$, $NO_3^-$, and $SO_4^{2-}$ at the AMS13 Oil Sands ground site for the observations and three model simulations. The hourly data were also studied, but are not shown in the time series.

We first note that the $NH_3$ VMRs in the measured time series are relatively low with mean, median, and maximum of 0.6 ppbv, 0.426 ppbv, and 2.98 ppbv, respectively in the hourly data, which are lower than the 1-8 ppbv range in Bytnerowicz et al. (2010), and the 2.7 ppbv summertime mean given in Hsu and Clair (2015). However, this may be due to the different time periods and locations measured. Our mean measured values at the AMS13 site are similar to the VMRs found at U.S. AMoN background sites (http://nadp.sws.uiuc.edu/amon/).

Figure 4a shows that the base model (green) background VMRs of $NH_3$ are very low (nearly 0 ppbv when there is no plume influence) compared to the measurements (orange). Only during the spike on September 3-4th does the base model exceed the measured values, probably indicating a local plume event fumigating to a lesser extent in the observations than was predicted by the model. The $NH_3$ VMRs of the base case are biased low compared to the surface measurements by a median of -0.35 ppbv (Fig. 5a) over the time period of the campaign. In Figure 4, the bidi model (blue line) and fire+bidi model (red line) show a significant improvement to the $NH_3$ VMRs compared to the base model (green line). Unfortunately, during some time periods, these two versions of the model overestimate $NH_3$: during August 13th, the model adds a significant level of $NH_3$ due to fire emissions, however the surface *in situ* observations show no evidence of fire impact. During other time periods (e.g., 30 August to 3 September, and 4-7 September), the bidi model appears to have put too much $NH_3$ into the system. Therefore, the bidi model bias (Fig. 5a) is now 0.30 ppbv too high (median), and the fire+bidi bias is 0.32 ppbv high (median) over the time period of the campaign, resulting in an overall improvement of only 0.03 ppbv in the model bias.

While the bias improvement is small, the bidi and fire+bidi both have greatly improved correlation coefficients (from R=0.1 to 0.4) and slopes much closer to 1 (from 0.1 to 0.7), showing that those added sources are important to improve model results (Fig. 6a). Additionally, the diurnal cycle (not shown) was improved in the bidi simulation, with both it and the measurements shaped like a sine curve with a minimum at 3:00-4:00am local time, and a maximum at noon local time, although the amplitude of the cycle was underestimated. Whereas, the base model diurnal cycle was flat from midnight to noon local time, and spikey from noon to midnight.

While Fig. 4a to 6a show that the addition of bidirectional flux improves the model correlation coefficient, slope, and bias, there is still room for further improvement. Paired t-test results indicate that the fire+bidi and measurements are still significantly different (see Table 2 for comparison statistics

of all three simulations). While inherent limitations from model resolution and uncertainties may be responsible for the remaining bias, it is likely that (a) the emission potentials for the land use categories (LUCs) in the region may be causing too much re-emission of $NH_3$, and need refinement, and (b) the fire emissions of $NH_3$ are not properly distributed in the vertical, placing too much $NH_3$ near the surface and/or the fire emission factors for $NH_3$ are too high.

Refinement needed for the emission potentials and LUCs may be a significant cause of the bidi and fire+bidi model biases. Rooney et al. (2012) have shown that about 64% of the AOSR are wetlands (fens, bogs and marshes), which should be mapped to the swamp LUC. However, our model currently assigns the AOSR landscape to evergreen needleleaf trees, deciduous broadleaf trees, inland lake, mixed shrubs, and mixed forests (and none of the region to swamp). This would lead to an overestimation of re-emission given that bogs are fairly acidic and our swamp emission potential is lower than the aforementioned LUCs. Other evidence for these two explanations will be presented below in Section 4.3.

The time series, model-vs-measured correlations, and model biases of $NH_4^+$, $NO_3^-$, and $SO_4^{2-}$ are also shown in Fig. 4 to 6 (b, c, and d, respectively). For $NH_4^+$ and $SO_4^{2-}$ there is very little change despite the increase in $NH_3$ that the bidirectional flux yields. The bias is very small for all three model scenarios, and the correlation coefficients are all relatively poor. So while there is an improvement to modelled $NH_3$ with bidirectional flux, there is a neutral affect on fine particulate $NH_4^+$. This may be because the charge of $NH_4^+$ in the particles is already enough in the base model to balance the charge of $2 \times SO_4^{2-} + NO_3^-$ in the aerosols, thus, causing any additional $NH_3$ (from bidi and fires) to remain in the gas phase. Alternatively, the minimal change in $NH_4^+$ could be due to additional wet scavenging of the additional $NH_3$, which will be discussed in Section 5.2. The change in $NH_3$ VMR has no effect on $SO_4^{2-}$ since particulate $SO_4^{2-}$ is not sensitive to the amount of $NH_3/NH_4^+$ available, and is dominated by anthropogenic and fire emissions. For $NO_3^-$, the base model bias was quite small at 0.01 $\mu g/m^3$, however the addition of bidi and fire+bidi further reduced that bias to 0.0011 and 0.0004 $\mu g/m^3$, respectively, which is a significant improvement. The correlation coefficient for $NO_3^-$ also improved from about 0.1 to 0.3 (Fig. 6c).

## 4.2 Along the OS campaign flight paths

There were 13 flights during the OS campaign that had valid (above detection limit, and no instrument error) $NH_3$ measurements, and 22 flights that had valid $NH_4^+$ (0-1 $\mu$m diameter) measurements. The flight path of the first flight, which occurred on 13 August 2013, is shown in Fig. 3; chosen as an example because this flight sampled mainly background $NH_3$ (rather than facility plumes).

Figure 7 shows the $NH_3$ VMRs along this flight path over time. Here the hourly model output is interpolated to the same time frequency as the measurements. The model output also has spatial resolution limits when comparing to the aircraft. However, we clearly see that for this flight, the bidirectional flux has increased $NH_3$ VMRs, bringing them closer to the measured values (median

biases for this flight are -1.38, 0.68, and 0.69 ppbv in the base, bidi, and fire+bidi simulations). There is little change when fires are added (Fig 7d vs c) because this flight did not pass through a fire plume.

Figure 8 shows the model-measurement differences and the model vs measurement scatter plots for the combined set of all flight paths for hourly-average $NH_3$ and $NH_4^+$. For $NH_3$ the median base model bias is -0.75 ppbv, comparable to the bias observed in Shephard et al., 2015, with the bidi model bias improving to -0.24 ppbv, and the fire+bidi bias to -0.23 ppbv. Also the best correlation coefficient and slope is achieved by the fire+bidi scenario. The use of the bidirectional flux has thus reduced the model bias relative to the aircraft observations by a factor of three. The fire+bidi simulation has the best statistics compared to measurements, as summarized in Table 2.

Again, the $NH_4^+$ results show little change despite the increase in $NH_3$. The small bias from the base case gets insignificantly smaller, and the slope and correlation coefficients are all negligibly changed.

### 4.3 In the vertical profiles across the region

The CrIS satellite has many observations over North America during the 2013 Oil Sands campaign. We have evaluated the model with these observations in a number of ways:

1. All daytime data from Aug 12 - September 7th, 2013; model-measurement comparisons over a large region encompassing Alberta and Saskatchewan, latitude range: 48-60 °N, longitude range: 100-122 °W), which contains agricultural areas, a number of cities, the northern boreal forest, and the Oil Sands facilities.

2. Case studies where we attempt to isolate fire emissions and non-fire conditions to evaluate both new components (fires and bidi) of the model.

The latitude and longitude ranges of our model-measurement pairs are given in Table 1. The satellite passes over these regions at approximately 1pm and 1am local time.

There were over 60,000 model-measurement pairs between the model and the CrIS satellite over the model domain during August 12th to September 7th, 2013. Figure 9a presents model biases for the entire dataset in a box and whiskers plot of the vertical $NH_3$ profiles at five vertical levels. The left-most panel (i) shows the $NH_3$ VMRs measured by CrIS, and the right-most panel (v) shows the diagonal elements of the CrIS averaging kernels, illustrating the sensitivity of the satellite measurements to each vertical level. The $NH_3$ VMRs over Alberta and Saskatchewan measured by CrIS are very similiar to those found by TES in the Shephard et al. (2015) study for the AOSR region.

The middle panels (Fig. 9a, ii-iv) show the model biases from the three simulations. The base model has a very similar bias to CrIS as the older version of GEM-MACH(v.1.5.1) had compared to TES observations in the Shephard et al. (2015) study – thus showing that the negative $NH_3$ biases were not improved with the use of the newer GEM-MACH version (v2) itself. The fire+bidi model has the smallest bias in the highest three layers, but the bidi model has the smallest bias in the two

lowest layers. In those lower layers, the fire+bidi model increases NH$_3$ VMRs too far (though still a smaller absolute bias compared to the base case, Fig. 9a). The fire+bidi positive bias could be due to an overestimate of the bidirectional flux re-emissions or of the fire emissions, or to an underestimate of the altitude of the fire emissions, or a combination of all three factors. In order to distinguish between these possibilities, two case studies were examined further in the next section. The statistics from the model-CrIS comparison can be found in Table 2. That summary shows that the fire+bidi simulation performs better than the base and the bidi simulations.

The spatial distribution of modelled NH$_3$ can also be evaluated with CrIS measurements, as shown in Figure 9b. These are maps of the average surface NH$_3$ from the base model, the fire+bidi model, and the CrIS satellite. The fire+bidi model over-predicts the effect of fires in the middle of northern Saskatchewan, but appears to be missing fires in north-western Manitoba. Other than fire influence, the spatial distribution in the fire+bidi model is the same as that of the base model, but with significant increases in overall VMR. The spatial distribution of the model simulations is different from the spatial distribution that CrIS measures. For example, the model predicts much higher NH$_3$ near the city of Edmonton than CrIS shows. That said, the addition of bidirectional flux has greatly improved the NH$_3$ simulation in the northern part of the province, where it was almost zero in the base model.

We selected three sample days (12 August and 1 and 3 September, 2013) that we use for the case studies. The measured surface NH$_3$ and sample Aqua MODIS true colour composite maps for those days are shown (Fig. 10). The four boxed regions on those maps indicate where model-measurement pairs were sampled for this study. The cyan and black boxes in Fig. 10a and b are the regions where we sample clear-sky, no-fire conditions on 3 and 1 September 2013, respectively. The magenta box in Fig. 10c is the region where we isolated our fire case study on 12 August 2013. The blue box is the region we discussed above, which we analysed for the full time period simulated (12 Aug - 7 Sep 2013, Fig. 9a).

### 4.3.1 Case study 1: clear-sky days with little fire influence - evaluating bidi

In order to evaluate the bidirectional flux component separately from the fire component, we selected September 3rd (northern, boreal forest and AOSR region - cyan box in Fig. 10a) and 1st (southern, agricultural region - black box in Fig. 10b), where the MODIS maps (EOSDIS NASA World view map, worldview.earthdata.nasa.gov) showed very little hot spots from fires, and conditions that were relatively cloud and smoke free (which yield the most CrIS observations). See Table 1 for the latitude and longitude ranges. Figure 10 also shows the surface NH$_3$ VMRs as observed by CrIS on each of those days. Figure 11a shows that in the north, the bidi model improves the bias from -0.84 ppbv to -0.07 ppbv in the lowest vertical level, and smaller, but still significant, improvements to the bias at the other levels. The fire+bidi model has a nearly identical impact as the bidi model, which is expected in a fire-free zone. Therefore, the GEM-MACH-Bidi model performs very well in northern Alberta and Saskatchewan where there is mainly boreal forest, and background-level NH$_3$. This also

implies that the LUC assignment discussed in Section 4.1 may only apply to a small region around the AOSR, and not to the overall large region we've defined here.

In the southern region (Fig. 11b), the addition of bidirectional flux moves the bias from near-zero to +1.02 ppbv in the lowest level. In this case, the base model with no bidirectional flux appears to be the most accurate model in areas dominated by agricultural sources. There are two possible explanations: a) agricultural emissions are too high in the base model, and the addition of the bidirectional flux leads to an overestimation of the $NH_3$ amounts, or b) re-emissions from bidirectional flux from crops are not significant. The literature (Bash et al., 2010; Massad et al., 2010; Zhang et al., 2010; Zhu et al., 2015) indicate that crops do indeed re-emit $NH_3$, therefore, (a) is the more likely explanation. The agriculture $NH_3$ emission inventory we used was created by the NAESI (National Agri-Environmental Standards Initiative) project (Bittman et al., 2008; Ayres et al., 2009; Makar et al., 2009), and has about 30-200% uncertainty associated with it (Bouwman et al., 1997; Asman et al., 1998). Therefore, with improved national $NH_3$ emission inventories, GEM-MACH-Bidi is likely to improve model results across the domain.

### 4.3.2 Case study 2: a clear day with significant fire influence - evaluating fires

In order to evaluate the fire component separately from the bidirectional flux, we selected August 12th (a northern region with little-to-no agricultural contributions) where the MODIS map shows numerous hot spots from fires and smokey conditions (Fig. 10c, magenta box). The base and bidi models underestimate $NH_3$ VMRs (Fig. 11c) by -6.22 and -5.84 ppbv, respectively (in the lowest vertical layer), but the fire+bidi model overestimates $NH_3$ by +4.06 ppbv. The fire+bidi version of the model still has the lowest bias of the three simulations, however, either (a) the fire+bidi model does not distribute the fire emissions properly in the vertical, (b) the fire emissions of $NH_3$ are too high, and/or (c) the model's oxidation rate of $NO_2$ and $SO_2$ in the fire may be underestimated, resulting in less sulfate and nitrate to convert $NH_3$ to $NH_4^+$. It is potentially a combination of all three explanations, as both fire plume rise and fire emission factors are on-going areas of study, and we further elaborate below.

Shinozuka et al. (2011) suggest that fire plumes are Gaussian-distributed in a thin layer aloft, which is not how our current fire emissions module distributes the fire plume. In our simulation, the fire emissions are distributed evenly throughout the boundary layer (the first 3-4 layers in Fig. 11c). However, we do not believe our parameterization of plume distribution causes the fire+bidi bias since the positive bias extends throughout the first three vertical layers and does not go negative in any layer (Fig. 11c) as would be expected if mass redistribution of the plume was the cause of the biases. We also know that the plume heights for most of the Fort McMurray fires of 2016 reached only up to 3-4 km altitude range based on the NASA Cloud-Aerosol Lidar and Infrared Pathfinder Satellite Observation (CALIPSO) and Multi-angle Imaging SpectroRadiometer (MISR) satellite observations. Therefore, the fire plumes are not located above the altitudes we studied.

Unfortunately, there were no flights that captured the fine structure of the fire plumes during the 2013 monitoring intensive campaign that can be used to further corroborate the vertical distribution of the fire plumes. There will however be flight observations of fires during the planned 2018 AOSR measurement campaign.

    Explanation (b) seems the most likely, as the uncertainty on emission factors for $NH_3$ from wild-
555 fires is very large (e.g., $\pm$50-100% depending on the fuel type, Urbanski, 2014), and could easily be overestimated. The $NO_x$ and $SO_2$ fire emission factors have smaller uncertainties of $\pm$10-40% (Urbanski, 2014). Therefore, the model may be further improved with reduced $NH_3$ emission factors for fires.

## 5   Impacts of bidirectional flux and forest fires on $NH_3$ VMRs

### 5.1   Effect on ambient ammonia

    Given that the overall fire+bidi model agrees best with measurements in the greater Alberta/Saskatchewan region (discussed throughout Section 4, and Table 2) and contains all known missing sources of $NH_3$, we can use the model to answer one of our key questions: What percent contributions to total ambient $NH_3$ VMRs came from bidirectional flux versus from forest fires during the study time period?
We do so by subtracting the bidi model output from the fire+bidi model output to get the forest fire component, and subtracting the base model output from the bidi model output to get the bidi component. The absolute differences are calculated as follows:

$$bidicomponent = NH_3^{bidi} - NH_3^{base} \tag{5}$$

$$firecomponent = NH_3^{fire+bidi} - NH_3^{bidi}, \tag{6}$$

which tell us how many ppbv of $NH_3$ on average is associated with re-emissions of $NH_3$ (upward component of bidirectional flux), versus fire emissions.

    The percent differences are calculated as follows:

$$bidipercent = \frac{NH_3^{bidi} - NH_3^{base}}{NH_3^{fire+bidi}} \times 100\% \tag{7}$$

$$firepercent = \frac{NH_3^{fire+bidi} - NH_3^{bidi}}{NH_3^{fire+bidi}} \times 100\%, \tag{8}$$

which tell us what percent of total $NH_3$ VMRs on average comes from re-emissions of $NH_3$ and from fire emissions, assuming the $NH_3$ from our fire+bidi simulation is the true total $NH_3$.

    We perform this calculation on the averaged model output (12 August to 7 September 2013) over
580 the 2.5-km model domain, and get an average of 20.3% (or 0.42 ppbv) and a median of 10.4% for ambient surface $NH_3$ VMRs that come from forest fires (Fig. 12b and d). The mean and median

are so different because fires are sporadic, large contributions to $NH_3$ VMRs, and the mean value is more sensitive to the big outliers. We get an average of 56.6% (or 1.24 ppbv) from bidirectional flux (56.3%, median, Fig. 12a and c), and the remaining 23.1%, average (33.3 %, median) comes from direct emissions from anthropogenic sources (agriculture, fossil fuel combustion, oil sands industry, etc). These numbers are summarized in Table 3. The increase in $NH_3$ due to the bidirectional flux scheme is of the same order of magnitude as that found in the Zhu et al. (2015) study using the GEOS-Chem model, during the month of July, in the United States (where they found 1 ppbv increase in surface VMRs due to bidirectional flux). It is also similar to values found in Europe in the Wichink Kruit et al. (2010) study.

Over the model domain, the minimum bidi influence on surface $NH_3$ is just north of Edmonton, where only 1% of $NH_3$ comes from bidi. Similarly, two AOSR facilities north of Fort McMurray stand out as having small bidi influence (12-40 %, surrounded by values in the 90s% - Fig. 12d). Also, any remote region with fire emissions will have a small percentage contribution from bidirectional flux during the fires, as they are in northern Saskatchewan (Fig. 12d). This is expected given that the average VMRs in cities and near large sources are very close to, or exceed the compensation point. The absolute maximum in the bidi component map is 4.5 ppbv in the lower right corner (an agricultural region with high $NH_3$ emissions), and the minimum is 0 ppbv (Fig. 12b). This means that nowhere in the domain, did the bidirectional flux formula result in more net deposition than the base model calculated via the Welesley/Robichaud/Zhang scheme. The maximum fire contribution to surface $NH_3$ is 27.9 ppbv where large fires occurred in northern Saskatchewan (Fig. 12c).

## 5.2 Effect on Deposition

Similar to our analysis from the previous section, we can use the model to determine how bidirectional flux and fires impact daily $NH_x$ deposition (which equals the dry deposition of $NH_3$ + the wet deposition of $NH_4^+$). Figure 13 shows the average daily net deposition (or net flux) of $NH_x$ from the base, bidi, and fire+bidi models. Negative (or blue) indicates net deposition (downward flux), and positive (or red), net emission (upward flux). The base model (Fig. 13a) had no re-emission (upward flux) option, thus $NH_3$ was always net dry deposited in that scenario. The bidi (Fig. 13b) and fire+bidi (Fig. 13c) maps show that most of the Alberta and Saskatchewan area has net deposition (e.g., near the cities, agriculture, and forest fires), but that some regions (with low atmospheric $NH_3$ VMRs) have net emission of $NH_x$. The dry $NH_3$ flux is net positive over the domain, however, when the increase in wet $NH_4^+$ is accounted for, the net flux of $NH_x$ is still negative (downward). This is very similar to what Wichink Kruit et al. (2010) found in a 2007 study in Europe; a reduction in dry $NH_3$, compensated by an increase wet $NH_4^+$ deposition.

Note that the assumption of an infinite soil pool of $NH_4^+$ in our bidirectional flux scheme has not caused an overwhelming upward flux of $NH_x$. In fact, the average results across the domain actually have more deposition in the fire+bidi scenario than in our base scenario. Table 4 shows the mean and

median net $NH_x$ flux for each scenario (presented as deposition, so negative signs removed). That said, following the soil pool approach (Pleim et al., 2013; Zhu et al., 2015), the soil pool of $NH_4^+$ may eventually get depleted. However, we believe this to be very unlikely for the following reasons: (1) Deposition of $NH_x$ throughout the year continually replenishes the soil pool – especially when temperatures are cooler in winter, spring, and fall, since the compensation point is exponentially dependent on temperature. (2) The short time frame of this study would not be long enough to deplete the soil pool. For example, Zhu et al. (2015) needed to spin up their model for three months in order to get the $NH_4$ soil pool stable, implying both a large pool, and a large time required for it to empty.

In the AOSR near Fort McMurray, we can compare our $NH_3$ dry deposition results to those calculated in Hsu et al. (2016). Their values range from 0.7 to 1.25 kg-N/ha/year (or 1.13 to $2.01 \times 10^{-5}$ moles/m$^2$/day), while ours are 10 times lower at around 0.13 kg-N/ha/year (scaled up to a year, from $2.12 \times 10^{-6}$ moles/m$^2$/day) near Fort McMurray, and do not vary much among our three model scenarios. These differences in deposition estimates are likely due to the fact that our study is only during a very warm time of the year, when deposition will be at a minimum, whereas the Hsu et al. (2016) study, covered both winter and summer time periods for multiple years. The differences may also be partially due to the fact that our modelled ambient $NH_3$ VMRs are also low compared to those measured in Hsu and Clair (2015) near Fort McMurray. They measured an average of $1.55 \pm 0.6$ ppbv (1.9 $\mu$g/m$^3$) at Fort McMurray, whereas our fire+bidi model has an average of 1.01 ppbv there (0.73 ppbv in bidi, and 0.39 in base). There may also be differences in that our model has more of the $NH_x$ deposition coming down as $NH_4^+$, rather than as $NH_3$.

Our fire+bidi $NH_x$ deposition values (Table 4 are well in line with reported $NH_3$ deposition in Kharol et al. (2017), who report satellite-derived $NH_3$ deposition of about 2.1-7.0$\times 10^{-5}$ moles/m$^2$/day in Alberta), and are at the low end of $NH_3$ deposition values reported within Behera et al. (2013).

The difference in deposition between the fire+bidi and bidi cases – which is the contribution of fires to the total $NH_x$ flux – showed that the fires increased downward flux/deposition over large swaths of the domain (e.g., difference between Figure 13c and b). The fires contributed an average of 1.954 $\times 10^{-5}$ moles/m$^2$/day of $NH_x$ deposition across the domain.

While the atmospheric concentrations of particulate $NH_4^+$ did not change much in our three simulations (see Sections 4.1 and 4.2), the wet deposition of $NH_4^+$ increased significantly going from the base to bidi to fire+bidi models. This is in contrast to what Zhu et al. (2015) found, which was little change to $NH_4^+$ wet deposition due to bidirectional flux. However that could be due other parameters, such as the meterological conditions, scavenging parameters, and/or gas-particle partitioning of $NH_x$. It would seem that in GEM-MACH-Bidi, the increased $NH_3$ is scavenged by precipitation. The average $NH_4^+$ deposition from the three simulations had a nearly threefold increase in the $NH_4^+$ deposition due to the increased $NH_3$ that the fire+bidi simulation yields. The average $NH_4^+$ wet deposition for our fire+bidi simulation is 5.86$\times 10^{-5}$ moles/m$^2$/day, which is in between values

reported in the United States in Stensland et al. (2000) (where they found an average of $1.9 \times 10^{-5}$ moles/m$^2$/day over the country), and in Japan in Murano et al. (1998) (where they found an average of $10.3 \times 10^{-5}$ moles/m$^2$/day over the country).

In the three scenarios, the average daily relative ratio of dry/wet deposition was: 0.43 for base, -0.77 for bidi, and -0.51 for fire+bidi (the negative value for the bidi and fire+bidi cases are because of

the average upward direction of NH$_3$). Since all average ratios are less than 1, this means that most of the removal process is from wet deposition, rather than dry deposition (even for the base case that had no re-emission of NH$_3$). Therefore, increased monitoring of wet deposition in the region would be useful. These results may also be useful for AEP terrestrial/aquatic scientists interested in nitrogen eutrophication. Maps of these ratios can be found in the supplemental material.

**6 Conclusions**

The GEM-MACHv2 air quality forecasting model was altered to include both the Zhang et al. (2010) bidirectional flux scheme for NH$_3$ and forest fire emissions of all species. This new "fire+bidi" model improves the simulated NH$_3$ in the modelled Oil Sands domain at 2.5-km resolution when compared to independent *in situ* measurements at the ground (at the AMS13 oil sands monitoring site) and aloft

(aircraft measurements), as well as at 10-km resolution when compared to cutting-edge satellite measurements from the CrIS instrument in Alberta and Saskatchewan. Almost all comparison statistics are best with our fire+bidi simulation. This suggests that the fire+bidi model shows promise for improving NH$_3$ model predictions elsewhere and during other time periods. However, more work is required to validate the model in other regions of the continent (e.g., with the Wood Buffalo En-

vironmental Association (WBEA) and the U.S. Ammonia Monitoring Network (AMoN) surface networks, and further CrIS satellite measurements), and for different time periods (e.g., springtime fertilizer season, cooler conditions, etc.). We have also shown that for further improvements in the Alberta/Saskatchewan region, the NH$_3$ emission factors for fires, and the NH$_3$ emissions from agriculture likely need to be reduced.

Despite the significant increase in atmospheric NH$_3$ VMRs with these additional sources, the impact on its byproduct, NH$_4^+$, was miniscule - as was the change to SO$_4^{2-}$ concentrations (0.02 $\mu$g/m$^3$ for each). The model bias for those species was not significantly changed in either direction. This is probably because of the extra NH$_4^+$ wet scavenging by precipitation, and the NH$_3$ VMRs were already high enough (before adding the extra sources) to charge balance the SO$_4^{2-}$ and NO$_3^-$ in

the aerosols. Thus, any additional NH$_3$ would remain in the gas phase. That said, the model bias for NO$_3^-$ at the AMS13 ground station was essentially removed with the fire+bidi model.

By running the base, bidi, and fire+bidi model scenarios, and taking the fire+bidi results as "true", we were able to calculate their differences and determine the average contributions from each source. We found that, on average, during the 12 August to 7 September 2013 time period in

the Alberta/Saskatchewan model domain, 23.1% of surface $NH_3$ comes from direct anthropogenic emissions, 56.6% of surface $NH_3$ comes from bidirectional flux (re-emission from soils and plants), and 20.3% of $NH_3$ comes from forest fires. Possible sources of error that remain in the bidi and fire+bidi simulations are the agricultural and fire emissions of $NH_3$, as well as the emission potentials for different land-use categories. The fraction of $NH_3$ from fires is highly variable depending on

the time periods and spatial domain analysed: on average from 12 August to 7 September 2013, the largest impact was in northern Saskatchewan. We also expect the re-emission source to be near the highest at this time of year because of the high temperatures, and this source should be much lower during the cold season, when deposition is expected to dominate the bidirectional flux process.

    The bidirectional flux process has decreased $NH_x$ deposition on average across the domain, with

some areas having a net emission of $NH_3$. However, that upward flux is due to the low atmospheric concentrations and high temperatures, and does not exceed the amount of $NH_x$ deposition that occurs during the cooler winter and spring times. When fires are also taken into account, the net $NH_x$ deposition is greater, on average across the domain, compared to the base model. The bidirectional flux process has decreased $NH_x$ deposition on average across the domain, with some areas having a

net emission of $NH_3$. However, that upward flux is due to the low atmospheric concentrations and high temperatures, and does not exceed the amount of $NH_x$ deposition that occurs during the cooler winter and spring times. When fires are also taken into account, the net $NH_x$ deposition is greater, on average across the domain, compared to the base model.

## 7    Data and code availability

Data availability: The CrIS-FRP-NH3 science data products used in this study can be made available on request (M. W. Shephard, ECCC). Similarly, the AMS13 observations can be made available on request from G. Wentworth (AEP). The aircraft observations are on the ECCC data portal (http://donnees.ec.gc.ca/data/air/monitor/ambient-air-quality-oil-sands-region/pollutant-transformation-summer-2013-aircraft-intens

    Model code availability: GEM-MACH - Atmospheric chemistry library for the GEM numerical

atmospheric model Copyright (C) 2007-2013 - Air Quality Research Division and National Prediction Operations division, Environment and Climate Change Canada. This library is free software which can be redistributed and/or modified under the terms of the GNU Lesser General Public License as published by the Free Software Foundation; either version 2.1 of the License, or any later version. Please contact the lead author (Cynthia Whaley, ECCC) for access to the GEM-MACH-Bidi

code, as there is currently no online link for download.

    Model data availability: Much of the emissions data used in our model is available on-line: Executive Summary, Joint oil sands monitoring program emissions inventory report (https://www.canada.ca/en/environment-climate-change/ and Joint Oil Sands Emissions Inventory Database (http://ec.gc.ca/data_donnees/SSB-OSM_Air/Air/Emissions_inventory_files/).

For the GEM-MACH-Bidi model output, please contact the lead author (Cynthia Whaley, ECCC) for hourly netCDF files of the 3-D ammonia fields.

*Acknowledgements.* The project was supported by ECCC's Oil Sands Monitoring program (OSM), and the Climate Change and Air Quality Program (CCAP). We would also like to acknowledge the University of Wisconsin-Madison Space Science and Engineering Center Atmosphere SIPS team sponsored under NASA contract NNG15HZ38C for providing us with the CrIS level 1 and 2 input data, in particular Keven Hrpcek and Liam Gumley.

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

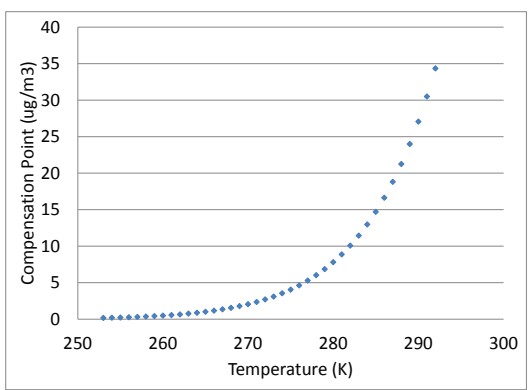

**Figure 1.** Compensation point ($C_g$) relationship to temperature; $C_g$ for evergreen needleleaf LUC shown as example.

**Table 1.** Latitude and longitude ranges that the model was evaluated over with the CrIS satellite measurements

| domain | date (in 2013) | lat range (°) | lon range (°) |
|---|---|---|---|
| AB/SK large domain | 12 Aug to 7 Sept | 48 to 60 N | -122.0 to -100.0 W |
| northern, no-fire case study | 3 Sept | 55 to 60 N | -120.0 to -110.0 W |
| southern, no-fire case study | 1 Sept | 49 to 53.5 N | -117.0 to -106.0 W |
| northern, fire case study | 12 Aug | 56.5 to 60 N | -110.0 to -104.4 W |

Zhang, J., Moran, M. D., Zheng, Q., Makar, P. A., Baratzadeh, P., Marsen, G., Liu, P., and Li, S.-M.: Emissions preparation and analysis for multiscale air quality modelling over the Athabasca oil sands region of Alberta, Canada, Atmos. Chem. Phys. Disc., submitted to ACPD Oil Sands special issue, 2017.

Zhang, L., Moran, M., Makar, P., Brook, J., and Gong, S.: Gaseous Dry Deposition in AURAMS A Unified Regional Air-quality Modelling System, Atmos. Environ., 36, 537–560, doi:10.1016/S1352-2310(01)00447-2, 2002.

Zhang, L., Brook, J. R., and Vet, R.: A revised parametrization for gaseous dry deposition in air-quality models, Atmos. Chem. Phys., 3, 2067–2082, doi:10.5194/acp-3-2067-2003, 2003.

Zhang, L., Wright, L. P., and Asman, W. A. H.: Bi-directional air-surface exchange of atmospheric ammonia: A review of measurements and a development of a big-leaf model for applications in regional-scale air-quality models, J. Geophys. Res., 115, D20 310, doi:10.1029/2009JD013589, 2010.

Zhu, L., Henze, D. K., Cady-Pereira, K. E., Shephard, M. W., Luo, M., Pinder, R. W., Bash, J. O., and Jeong, G.-R.: Constraining U.S. ammonia emissions using TES remote sensing observations and the GEOS-Chem adjoint model, J. Geophys. Res., 118, 3355–3368, doi:10.1002/jgrd.50166, 2013.

Zhu, L., Henze, D., Bash, J., Jeong, G.-R., Cady-Pereira, K., Shephard, M., Luo, M., Paulot, F., and Capps, S.: Global evaluation of ammonia bidirectional exchange and livestock diurnal variation schemes, Atmos. Chem. Phys., 15, 12 823–12 843, doi:10.5194/acp-15-12823-2015, 2015.

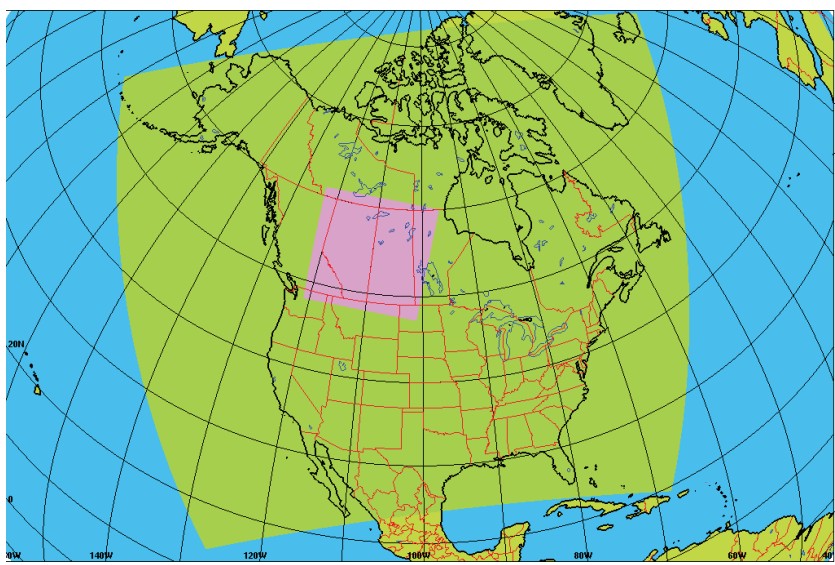

**Figure 2.** Map of 10-km resolution continental piloting model domain (green), and 2.5-km resolution nested model domain (purple).

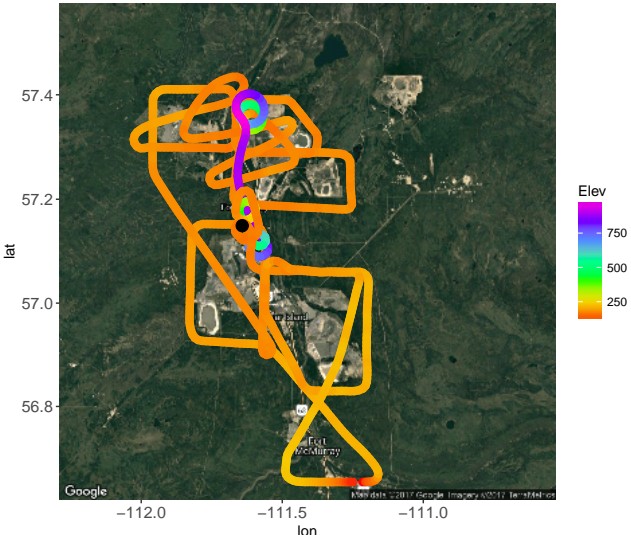

**Figure 3.** Flight path on 13 August 2013, where elevation (in meters) is denoted by the colour scale, and the AMS13 site is indicated by a black circle.

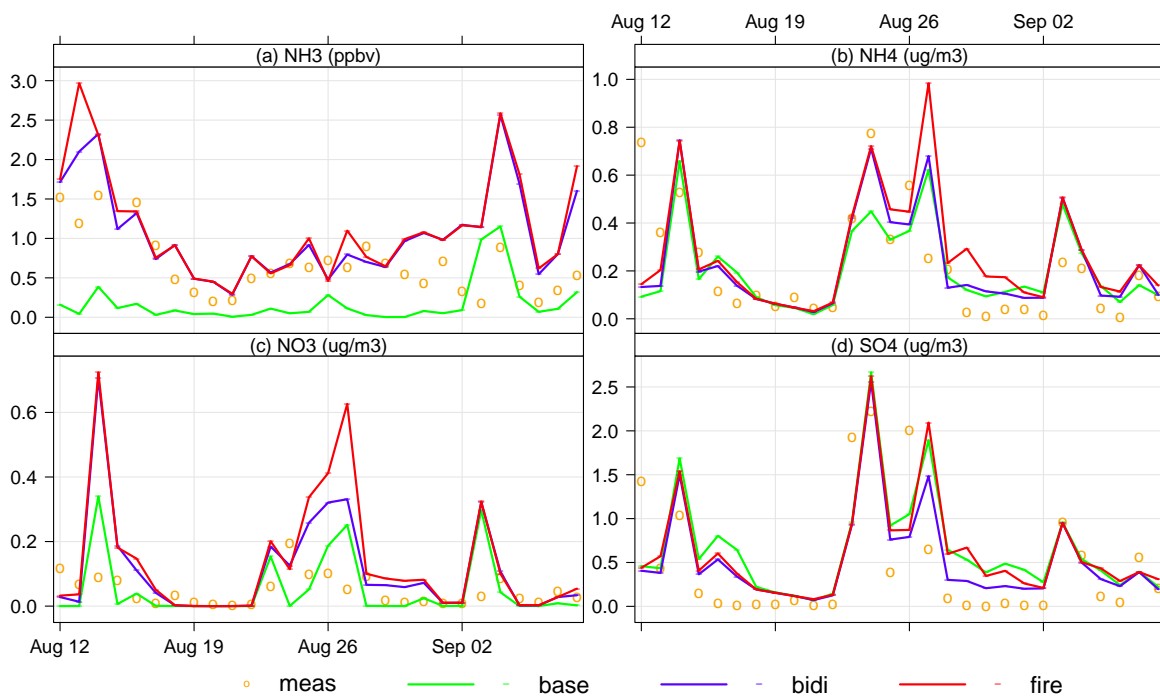

**Figure 4.** Surface daily average VMR of (a) $NH_3$, and concentrations of (b) fine particulate $NH_4^+$, (c) $NO_3^-$, and (d) $SO_4^{2-}$ at the AMS13 ground site in the AOSR. Measurements in orange, base model in green, bidirectional flux model in blue, and fire+bidi model in red.

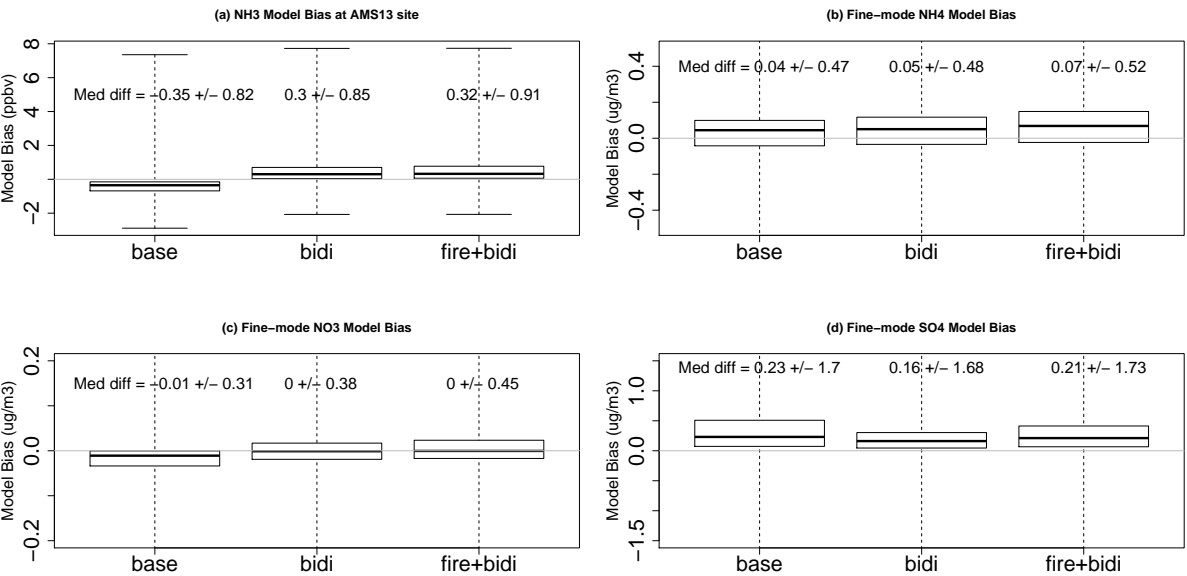

**Figure 5.** Hourly model-measurement bias in surface (a) NH$_3$ VMR, and (b) NH$_4^+$, (c) NO$_3^-$ and (d) SO$_4^{2-}$ concentrations at the AMS13 ground site in the AOSR.

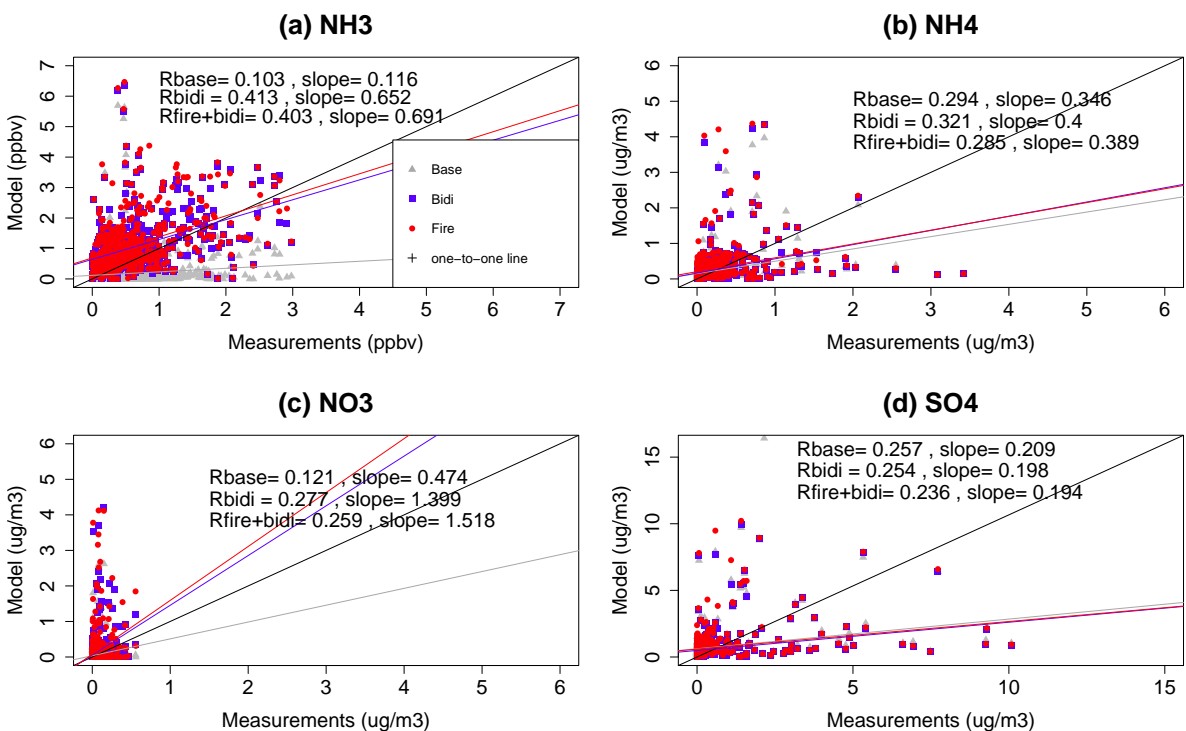

**Figure 6.** Hourly modelled vs measured surface (a) $NH_3$ VMR, and (b) $NH_4^+$, (c) $NO_3^-$ and (d) $SO_4^{2-}$ concentrations at the AMS13 ground site in the AOSR. Base model in grey, bidirectional flux model in blue, and fire+bidi model in red.

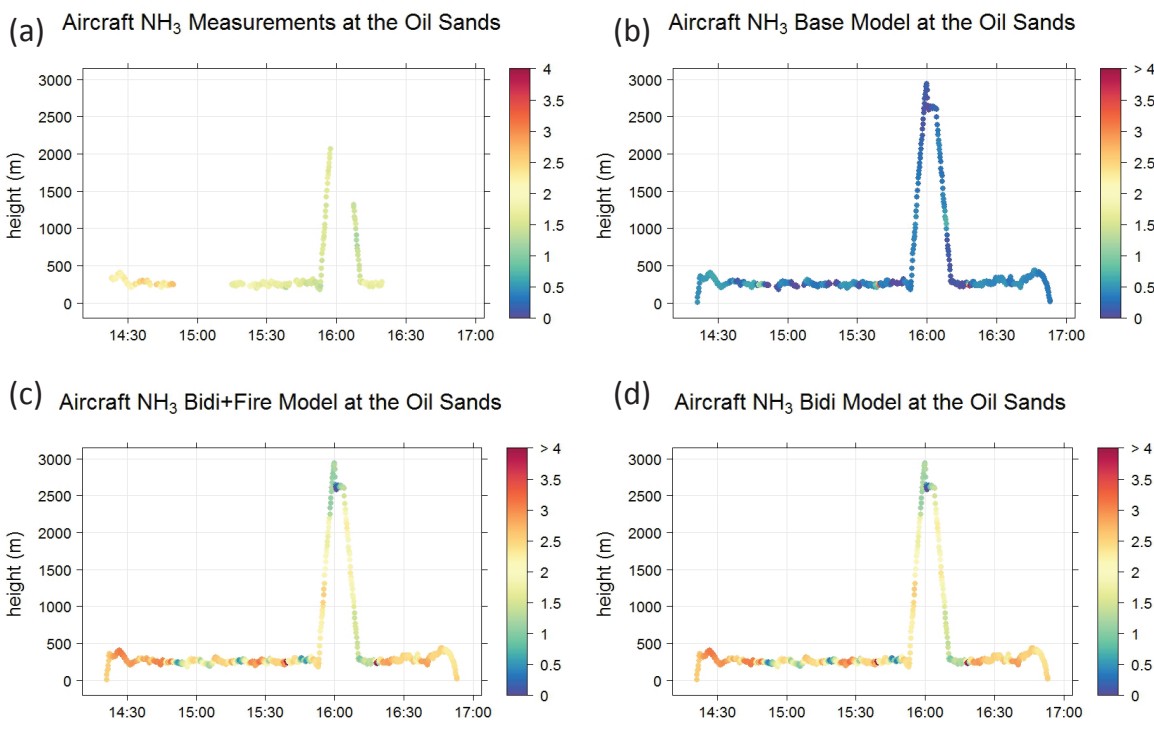

**Figure 7.** NH$_3$ VMRs aloft (colour scale) over the OS region during the 13 August 2013 flight. (a) measurements, (b) base model, (c) fire+bidi model, and (d) bidi model.

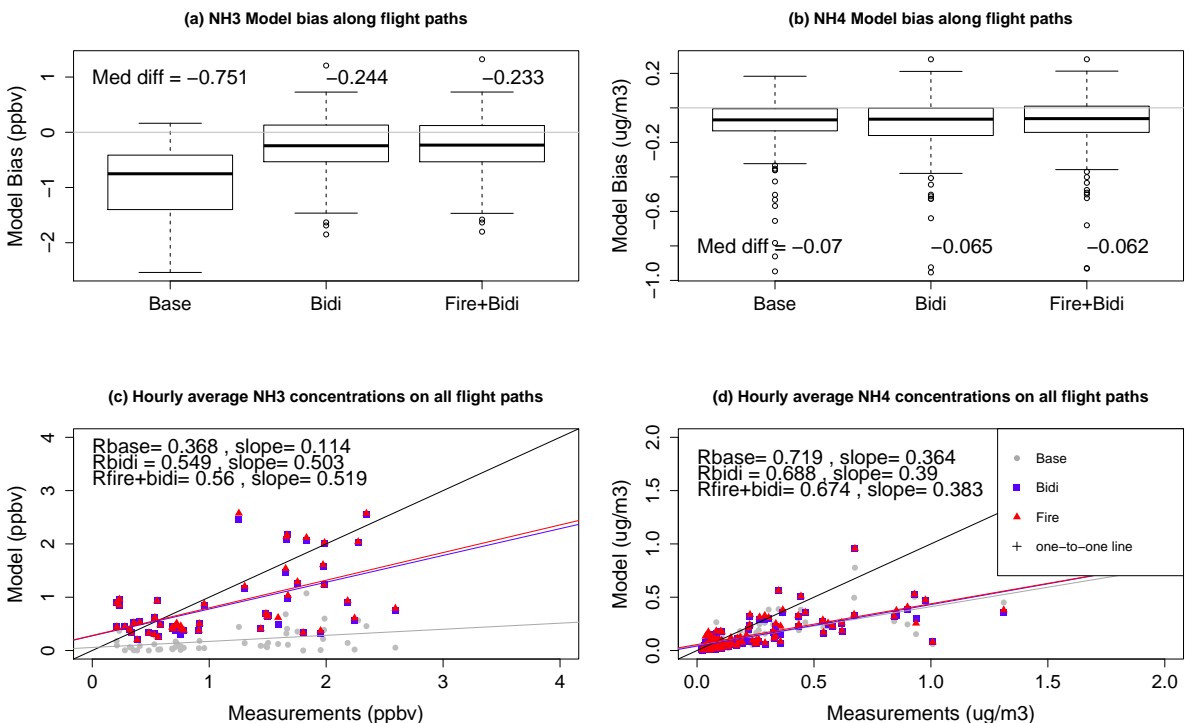

**Figure 8.** Hourly averages along all flight paths over the OS region during the summer 2013 campaign: Model-measurement bias in (a) $NH_3$ and (b) $NH_4^+$. Modelled vs measured (c) $NH_3$ VMR and (d) $NH_4^+$ concentrations aloft. Base model in grey, bidirectional flux model in blue, and fire+bidi model in red.

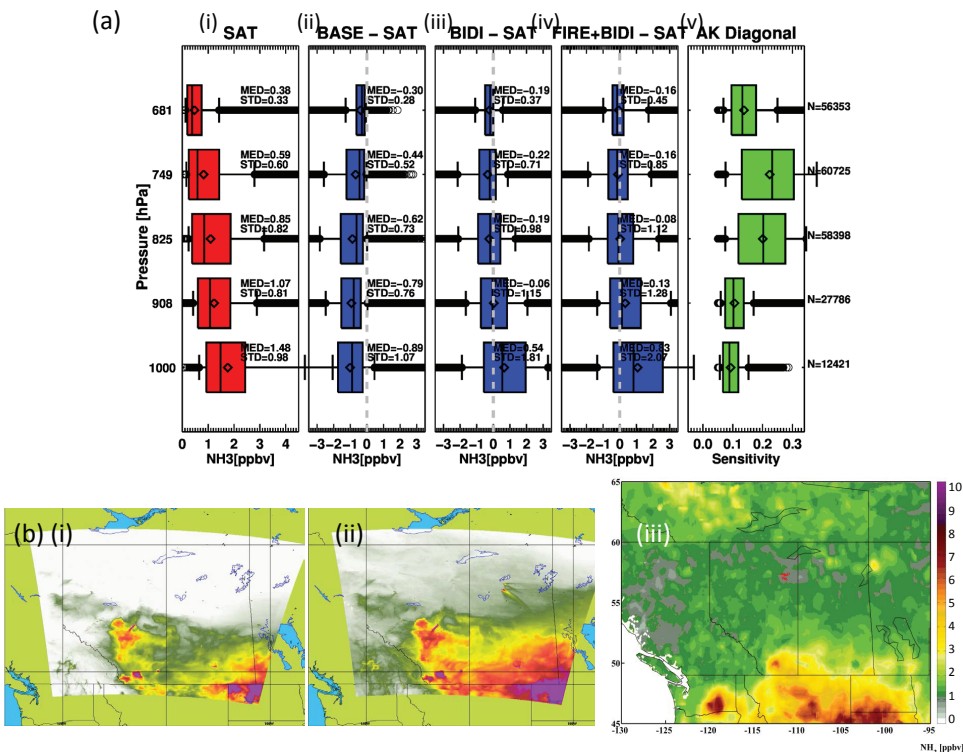

**Figure 9.** (a) (i) NH$_3$ vertical profiles as measured by CrIS satellite from 12 Aug to 7 Sep 2013; difference between measurement and (ii) base model, (iii) bidi model, and (iv) fire+bidi model; and (v) averaging kernel of CrIS satellite for NH$_3$ retrieval. (b) Average (12 Aug - 7 Sep 2013) surface NH$_3$ VMRs given by the (i) base model, (ii) fire+bidi model, and (iii) CrIS satellite.

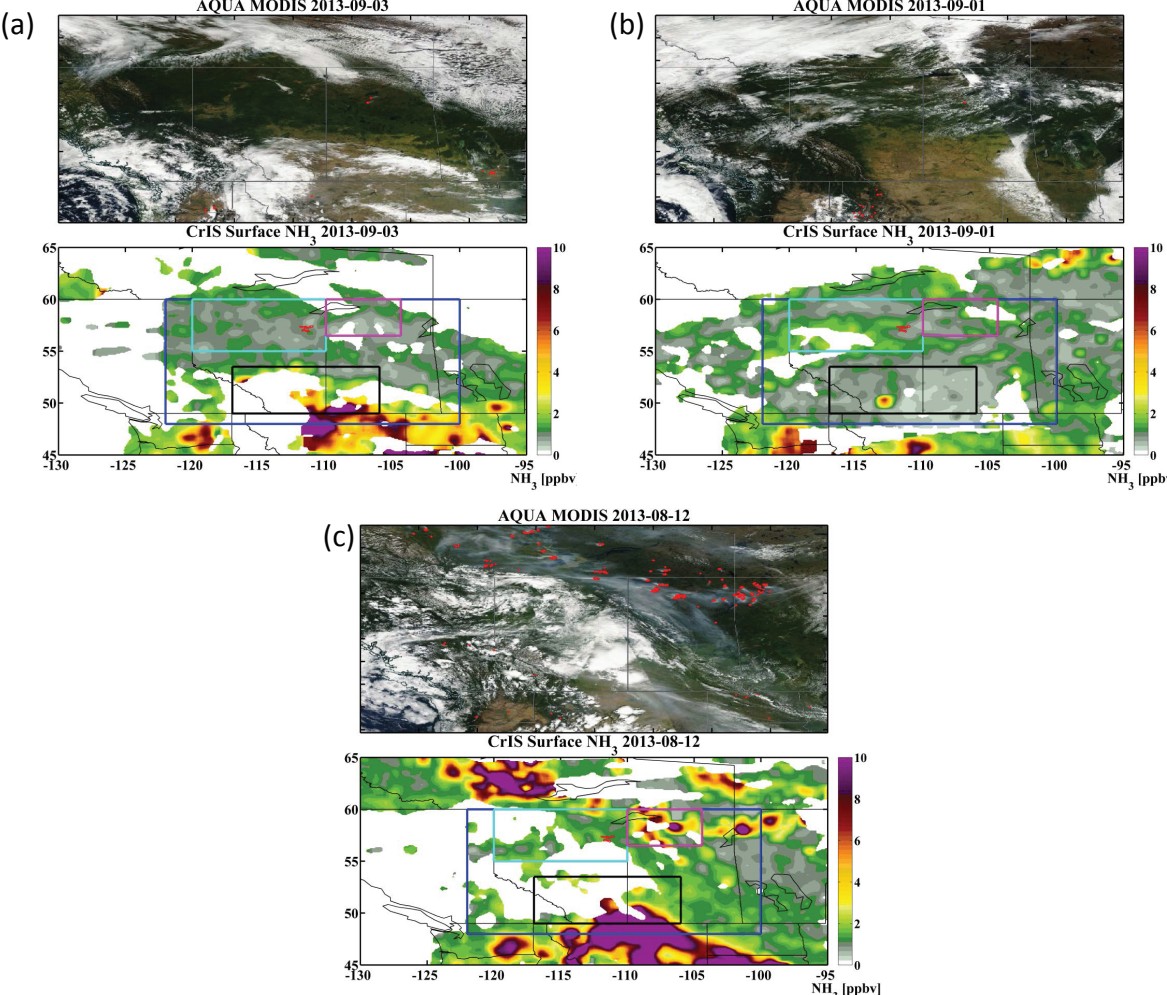

**Figure 10.** (Top panels) Images of the Alberta/Saskatchewan region with clouds and fire hotspots from MODIS. (Bottom panels) Map of CrIS-measured surface NH₃ VMRs, with coloured boxes showing the regions where model and satellite measurements were sampled. These three examples are for (a) northern bidi case study (cyan), (b) southern bidi case study (black), and (c) fire case study (magenta), discussed in Section 4.3), and the blue box is the region of our overall comparison.

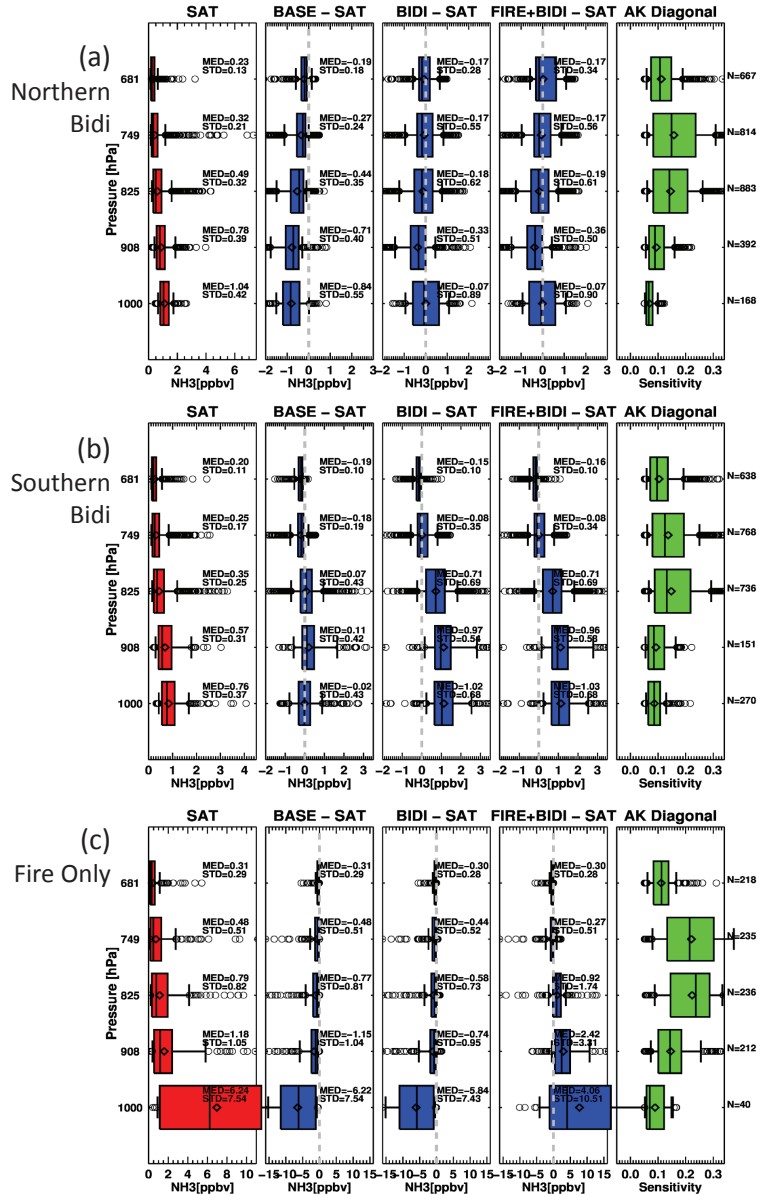

**Figure 11.** As in Fig. 9, but for our (a) northern "bidi-only" case study (3 Sept 2013), (b) southern "bidi-only" case study (1 Sept 2013), and (c) northern "fire-only" case study (12 Aug 2013). Regions are shown in Figure 10a (cyan), 10b (black), and 10c (magenta) boxes, respectively.

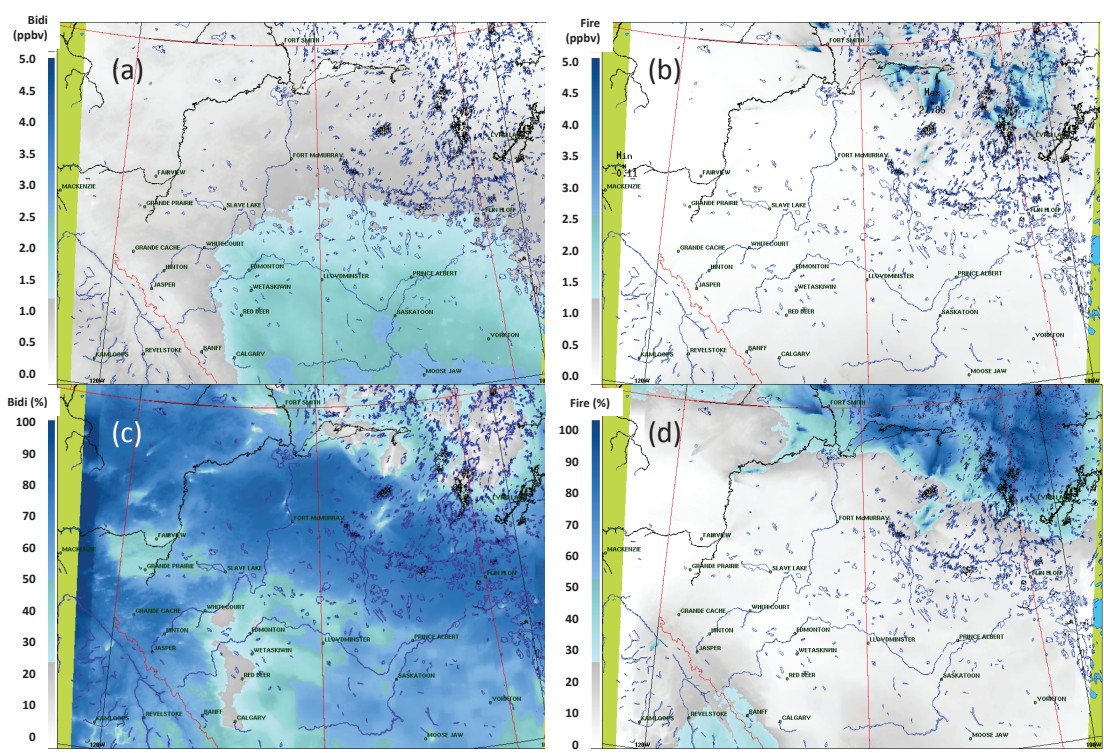

**Figure 12.** Maps of the modelled (a) absolute bidirectional flux contribution, (b) absolute fire contribution, (c) percent bidirectional flux contribution, and (d) percent fire contribution to surface NH$_3$. These are averages over 12 August to 7 September, 2013.

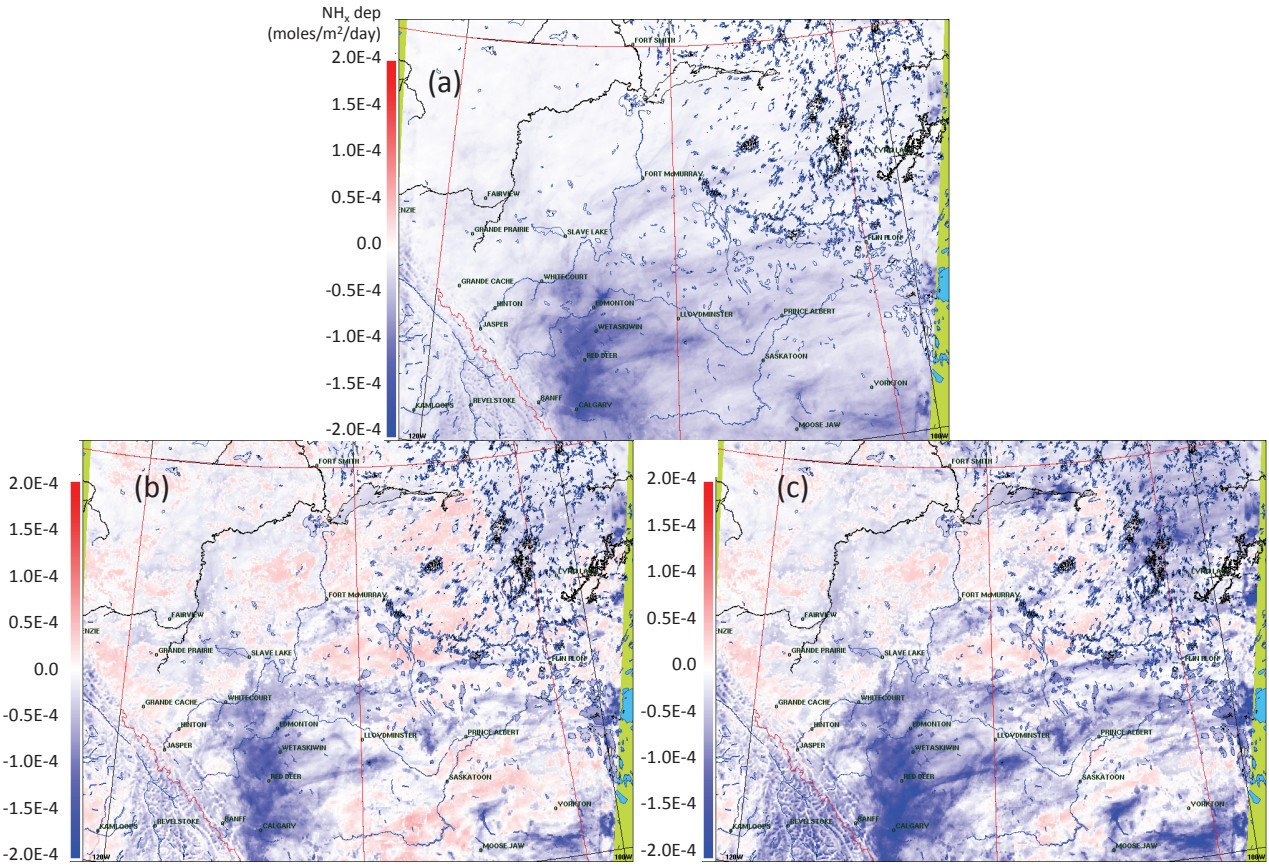

**Figure 13.** Maps of the modelled average NH$_x$ deposition for (a) base (b) bidi, and (c) fire+bidi models. In all maps, red/positive represents upward flux, and blue/negative represents downward flux. These are daily amounts, averaged over 12 August to 7 September, 2013.

**Table 2.** Model-measurement $NH_3$ comparison statistics from 12 August to 7 September 2013: R=correlation coefficient; slope is of the line-of-best fit between model vs. measurement; p and t are from a paired t-test between model and measurement data pairs (p>0.05 and |t|<1 means that the model is statistically indistinguishable from measurements); the median model bias; RMSE=root-mean-square error; and FE=fractional error of the models. CrIS (troposphere) results are for the entire model domain at all tropospheric levels shown in Figure 9(top), and CrIS (surface) results are for the lowest retrieval level (both are during mid-day satellite overpass times); aircraft results are from the 12 flight paths over the oil sands facilities, hourly averages during the daytime; and AMS13 results are from hourly data (day and night) at the one ground station.

| | R | slope | p | t | bias (ppbv) | RMSE (ppbv) | FE |
|---|---|---|---|---|---|---|---|
| CrIS (troposphere) | | | | | | | |
| base | 0.248 | 0.076 | <2E-16 | -247.5 | -0.430 | 2.02 | -5.3E-6 |
| bidi | 0.302 | 0.205 | <2E-16 | -77.4 | -0.176 | 1.93 | -1.2E-6 |
| fire+bidi | 0.338 | 0.425 | <2E-16 | 36.2 | -0.126 | 2.45 | 5.9E-7 |
| CrIS (surface) | | | | | | | |
| base | 0.272 | 0.118 | <2E-16 | -19.0 | -1.11 | 5.72 | -1.6E-3 |
| bidi | 0.289 | 0.162 | <2E-16 | -12.8 | -0.66 | 5.32 | -8.9E-4 |
| fire+bidi | 0.566 | 1.195 | 1.4E-06 | 4.9 | -0.19 | 8.67 | 3.7E-4 |
| aircraft (hourly) | | | | | | | |
| base | 0.368 | 0.114 | 8.5E-14 | -10.3 | -0.751 | 1.14 | -2.5E-3 |
| bidi | 0.549 | 0.503 | 0.0026 | -3.2 | -0.244 | 0.69 | -5.0E-4 |
| fire+bidi | 0.560 | 0.519 | 0.0052 | -2.9 | -0.233 | 0.68 | -4.5E-4 |
| AMS13 (hourly) | | | | | | | |
| base | 0.103 | 0.116 | <2E-16 | -12.4 | -0.35 | 0.92 | -1.6E-3 |
| bidi | 0.413 | 0.652 | <2E-16 | 12.1 | -0.30 | 0.95 | 8.0E-4 |
| fire+bidi | 0.403 | 0.691 | <2E-16 | 13.1 | 0.32 | 1.04 | 9.0E-4 |

**Table 3.** Average source contributions to ambient $NH_3$ VMRs over the AB/SK model domain during 12 Aug to 7 Sep 2013.

| source | median (ppbv) | median (%) | average (ppbv) | average (%) |
|---|---|---|---|---|
| total surface $NH_3$ | 1.60 | 100 | 2.53 | 100 |
| from fires to surface | 0.25 | 10.4 | 0.42 | 20.3 |
| from bidi to surface | 0.97 | 56.3 | 1.24 | 56.6 |
| from anthro to surface | 0.38 | 33.3 | 0.87 | 23.1 |
| total column $NH_3$ | 18.8 | 100 | 25.6 | 100 |
| from fires to total column | 6.1 | 27.7 | 8.1 | 30.5 |
| from bidi to total column | 8.8 | 48.1 | 11.15 | 50.0 |
| from anthro to total column | 3.9 | 24.2 | 6.35 | 19.5 |

**Table 4.** Average $NH_x$ deposition (downward flux) over the AB/SK model domain during 12 Aug to 7 Sep 2013.

| Net Flux (moles/m$^2$/day) | base | bidi | fire+bidi |
|---|---|---|---|
| mean | $3.025 \times 10^{-5}$ | $1.811 \times 10^{-5}$ | $3.765 \times 10^{-5}$ |
| median | $2.061 \times 10^{-5}$ | $1.299 \times 10^{-5}$ | $2.843 \times 10^{-5}$ |