# Peer review of "Contributions of natural and anthropogenic sources to ambient ammonia in the Athabasca Oil Sands and north-western Canada"

_Atmospheric Chemistry and Physics, 2017_

## Referee Comment (RC1) · Anonymous Referee #1 · 2 Nov 2017

Review on acp-2017-627

Contributions of natural and anthropogenic sources to ambient ammonia in the Athabasca Oil Sands and north-western Canada By Cynthia Whaley, Paul A. Makar, Mark W. Shephard, Leiming Zhang, Junhua Zhang, Qiong Zheng, Ayodeji Akingunola, Gregory R. Wentworth, Jennifer G. Murphy, Shailesh K. Kharol, and Karen E. Cady-Pereira

The topic of the paper is about improving modeling representation of NH3 concentrations in the Alberta Oil Sands region, by running the GEM-MACH model with new parameterizations for bidirectional exchange of NH3 and NH3 emissions from fires.

[Figure]

The results are based on previous simulations proposed by Shepard et al. (2015) and are compared to surface, aircraft and satellite measurements.

General comments:

The subject of study (evaluating the model capability to reproduce NH3 concentrations) is interesting, and the paper is easy to follow. The structure is well organized. Some sentences need rephrasing, and some explanations are confuse (see specific comments).

Citations of non available papers are given (cited as "this issue" but non available in the special issue). This stands for example for description of measurements (Wentworth et al), deposition of N and S (Makar et al), parameterizations (Akingunola et al), fire emissions (Zhang et al.).

NH3 concentrations are really low in the region of study and the reader may wonder why this region is worth studying, in the light of what is written in the introduction, (NH3 may be harmful for air, water quality, or ecosystem and human health). A sentence or 2 on the relevancy of studying regions where concentrations remain low for the moment would be useful.

Some citations are given for satellite measurements of NH3 concentrations at a large scale, but nothing is said about orders of magnitude. Do the model results of this study match with previous satellite measurements? More values of ambient concentrations should be cited. This is also true for bidirectional exchanges: some papers are cited, but only as a list of papers, and no quantified values are given to be compared with what is found in this study. The papers should be cited with precise examples of measured fluxes is the same area or in regions with the same type of ecosystems.

A meteorological description of the site would be useful all along the study. Indeed, emissions have their impact on NH3 concentrations, but wind speed, humidity, temperature have also a significant impact on exchange fluxes. This part of dynamical

**[ACPD](ACPD)**
interpretation on deposition fluxes is missing. A discussion about why the addition of bidirectional flux is so important in improving the model is missing in terms of processes. The discussion is only about ppb and %, and not about processes.

1- Does the paper address relevant scientific questions within the scope of ACP?: yes 2- Does the paper present novel concepts, ideas, tools, or data?: not really, bidirectional exchanges of NH3 is already known to be important, tools and data have already been used in Shepard et al (2015) 3- Are substantial conclusions reached?: conclusions are reached 4- Are the scientific methods and assumptions valid and clearly outlined? yes 5- Are the results sufficient to support the interpretations and conclusions? yes 6- Is the description of experiments and calculations sufficiently complete and precise to allow their reproduction by fellow scientists (traceability of results)? 7- Do the authors give proper credit to related work and clearly indicate their own new/original contribution? yes 8- Does the title clearly reflect the contents of the paper? yes 9- Does the abstract provide a concise and complete summary? yes 10- Is the overall presentation well structured and clear? yes 11- Is the language fluent and precise? Not always 12- Are mathematical formulae, symbols, abbreviations, and units correctly defined and used? yes 13- Should any parts of the paper (text, formulae, figures, tables) be clarified, reduced, combined, or eliminated? Yes, mentioned in specific comments 14- Are the number and quality of references appropriate? Not always

Specific comments

Line 42: As you write that NH3 is a contaminant, precise in what order of concentration it has negative effects.

Precise somewhere in your introduction that despite negative effects of high concentrations, low concentration regions are also worth studying.

Line 44: Modeling provides. . .: this could be true if inventories are correct and with fine resolution, which is hardly feasible in most models. Remove this first part of the sentence.

Line 55: reformulate your sentence because reading "the AOSR is a large source of air" is a bit weird.

Line 58: why are NH3 concentrations so low, despite local pollution?

Line 61: give mean concentrations of cited agricultural areas.

Line 66: give values of the fraction of deposited NH3 compared to NO2 and HNO3.

Line 114: precise what species are used from the inventories.

Line 121: A word about the importance of carefully design stack parameters would be useful to understand why this part is so important for your study.

Line 167: give reference values of realistic NH3 concentrations consistent with findings from the literature, and explain why you use low end values.

Line 191: what do you mean by "major point emissions?"

Line 220: what is ECCC?

Line 259: you mention +/- 15% of uncertainty for measurements, with a mean measured value of 0.63 ppb (line 59). The model/measurements bias is 0.3±0.85 for bidi, what is the real impact on modeled concentration? What is the range of possible concentration? Is the measured concentration included in this range?

Line 299: Figure 3 should not be placed here. It is only used later in the text, and should be included before figure 9. Furthermore, if placed here, it is not understandable why these three dates are chosen (not explained in the caption).

Line 324: You mention fig 5 and then you talk about fig 4. Place fig 5 after fig 4 and give a description on interpretation of it.

Line 326: what are the background times?

In figure 4, NH3 is in ppb, but NH4+, NO3- and SO42- are not in ppb. Please correct the caption
Line 335: what is the increase of concentration with the influence of a local plume? 0.5 ppb seems to be very very low, and more included in the measurement noise than in a local pollution signal.

Line 337: 0.08 ppb is less than the 15% measurement uncertainty. Is it really significant?

Line 339: R=0.2 and 0.4. Are these coefficients significant? Could you give a significance (p-value) of your correlation calculation everywhere it is necessary?

Line 340: Fig 6a: what is the unit of NH3 concentration?

Line 368: SO42- is influenced by anthropogenic emissions, why not by fire emissions?

Line 375: Why did you choose this precise flight? By the way, it would be useful to give some average values of meteorological conditions when describing the area of study (mean temperature, humidity, rainfall, wind speed, etc... all parameters that have a possible influence on NH3 concentration)

Line 376: can you explain why fire+bidi does not improve the results compared to bidi?

Line 404: Figure 3 is used in this paragraph. It should appear in the text at this time, and not before.

Line 430: specify at the lowest level.

Line 446: remove "that" before the fire+bidi model

Line 451: You suppose that the conversion of NH3 to NH4+ is underestimated: did you have a llook et the NH4+ pool in that case?

Line 453 to 459: this explanation is very confused. Please rephrase.

Line 507: this sentence is not useful. Obviously if bidirectional parameterization is used it will balance deposition with emission and not increase deposition fluxes

Line 517 and elsewhere: the unit for the flux is not appropriate. Please homogenize

throughout the paragraph and use preferably ngN.m-2.s-1. Lines 584 and 587 use another unit which is not a flux unit. In this paragraph needs bibliography values need to be included for equivalent ecosystem or region. Figure 12 and 13 are redundant. Figure 13 is not necessary in my opinion.

Line 539: again this sentence is not useful. Obviously wet deposition is only deposition. Again flux units are not correct and should be homogenized.

Line 550: how do wet deposition fluxes compare with literature?

Line 570-571: this sentence has already been written above. The conclusion should mention the possible influence of meteorological conditions on NH3 concentration, as well as in the text.

Line 574: "miniscule": please quantify.

Technical comments

Line 16: remove , aftertime period.

Line 108: write covers instead of covering, twice in the line.

Line 109: remove "And" at the end of the line.

Line 117: remove ( before Zhang and put ( before 2017.

Line 163: replace "in" by "from" before Wen et al.

Line 228: Time period "from" . . .

Line 296: a verb is missing in the sentence.

Line 321: include "to" after compared.

Line 352: remove italics for "should", same line 488 for "total" and line 507 for "more"

Line 454: a ) is missing after 10c

[Figure]

Line 493: problem with the sentence, please rephrase

Line 507 and 509: remove , after 11.

Line 526: replace $\times$ by times

---

## Referee Comment (RC2) · Anonymous Referee #3 · 14 Nov 2017

In this work, the authors evaluate the GEM-MACH chemical transport model simulation of ammonia and other compounds in the Canadian oil sands region and assess the changes in model performance with the incorporation of ammonia bidirectional flux and wildfire ammonia emissions against ground-based, aircraft, and satellite data for the time period August 12 to September 7, 2013. They then use the model to assess the contributions of natural and anthropogenic emissions to ambient NH3 under the assumption that all reemitted NH3 was natural.

The manuscript is generally well written and easy to understand. Improving our ability to simulate NH3 and NH4 (NHx) is an important issue, and the incorporation of bidi-

rectional NH3 flux in chemical transport models is a needed and still emerging issue, which makes this work potentially important. However, I have a serious concern. In the authors' implementation of the bidirectional NH3 flux, they assumed that there was an infinite soil pool of NH4+. This is an unreasonable assumption that is recognized and discussed by the authors. However, due to this or other assumptions in the implementation of the NH3 bidirectional flux mechanism, the NH3 emission/reemission flux is similar to or greater than the total (wet + dry) NHx deposition. This implies that the ecosystems are taking up little to no deposited NHx, which does not seem to be a reasonable result during the growing season. This casts doubt that any improvements in model performance is for the "right reasons" and on the value of the source apportionment results. I think that the authors should investigate and discuss the net total reduced nitrogen deposition, and if they cannot justify the high emission/reemission rates of ammonia, then I question the value of the final source attribution results.

The authors pursued the incorporation of ammonia bidirectional flux and wildfire emissions into the model due to significant underestimations of ammonia concentrations in a previous modeling exercise. While reasonable, they do not discuss potential issues with other modeling inputs and processes, including the underestimation of emissions from other sectors, e.g., agricultural regions and NH3 slip in fossil fuel combustion systems, as well as potentially overestimating NH4 wet deposition. Early in the manuscript it would be good to discuss why these other factors are not likely significant contributors to the initial model underestimation. This could include evaluation of the model NH4 wet deposition simulation against measured wet deposition or through fall data. If NHx wet deposition is also underestimated, then this would certainly point toward biases in the dry deposition rates and/or emissions. Near the end of the manuscript, the authors do show that the base-case model simulation performed well near agricultural activity and that it underestimated NH3 when wildfire emissions impacted the area. This information supports the authors' premises, and I suggest that these results be discussed before the model comparison to the surface and aircraft measurements. Last, the oil sands region is an area of intense energy development, and some discussion of the

ammonia emission from this activity and its uncertainty is warranted.

Specific comments In the abstract and introduction it is noted that the Alberta oil sands region has relatively low ammonia concentrations. Please put this into some context. These concentrations are not low compared to many rural western North American sites. Also, can anything be said about the estimated deposition rates in these regions compared to the reactive nitrogen critical loads? If the deposition rates are near or above the critical loads, then this work could have important policy implications.

Lines 173-175: "the bidirectional flux acts effectively as an additional source of NH3 gas, releasing stored NH3 until and unless the ambient concentration rises to the compensation point concentration." It would be good to discuss the origin of the NH3 in these emissions. That is, is the NH3 originating from the natural processes of the ecosystem or from previously deposited NH3 or a combination of both? Presumably, it is from both. This also has implications when discussing natural versus anthropogenic NH3. The authors assume that all NH3 bidirectional flux emissions are natural; however, if the deposited NH3 originating from anthropogenic sources was reemitted, then this NH3 would have anthropogenic origins. Consequently, not all of the reemitted ammonia due to the bidirectional flux processes is necessarily natural.

Line 186: "it is not desirable for our bidirectional flux scheme to have to rely in advance on another model's output. Therefore, we use this simplified version, and assess whether its results provide a good enough improvement to simulated NH3 for less cost in run time." The authors did not discuss what constitutes a "good enough" model simulation or whether the studied model satisfied this criterion. In addition, as discussed in the general comments, I question whether the high NH3 emissions resulting from the bidirectional flux mechanism are reasonable or not and suggest further investigation and discussion.

Line 307: "Figure 4 shows the time series of the concentrations of NH3 and its reaction products, fine-particulate NH4+ and NO3-..." This is a confusing sentence. Also

please specify if NO3- is only particulate nitrate or if it includes nitric acid.

Section 4.1: Reproducing the measured hourly ammonia concentrations is very challenging. It would be good to see how the model performs on an aggregated basis as well, e.g., can it reproduce the 24-hour average NH3 values and the average diurnal cycles?

It would be good to include estimates of the model error such as the RMS and fractional errors and bias in the model performance statistics.

Line 333: "(from R=0.2 to 0.4)..." From Figure 6 it looks like the improvement in correlation should be from 0.1 to 0.4.

Line 372: "However, we clearly see that for this flight, the bidirectional flux has increased NH3 concentrations, bringing them closer to the measured values." It is not clear from the figure that the model performance has improved, only that the simulated NH3 has increased. It would be good to add performance stats to panels b–c in Figure 7.

Lines 425-435: I think this discussion is very important for justifying the modeling refinements and should be moved up front.

Technical comments The fonts used in the figures are very small, making text difficult to read. This is particularly the case in Figures 3, 11, 12, 13, and 14 and supplemental material.

Figures 12 and 13 are missing panels.

---

## Author Comment (AC1) · 28 Nov 2017

Thank you for your thorough review of our paper. We will be taking into account all of your suggestions, which will greatly improve the manuscript. As part of the interactive ACPD discussion, we would like to address your major concern early on. Please see below for our response to reviewer #3's major concern (reviewer comment in italic font), and the rest of our responses to all reviewers will follow at a later time.

*Serious concern: In the authors' implementation of the bidirectional NH3 flux, they assumed that there was an infinite soil pool of NH4+. This is an unreasonable assumption that is recognized and discussed by the authors. However, due to this or other assumptions in the implementation of the NH3 bidirectional flux mechanism, the NH3 emission/reemission flux is similar to or greater than the total (wet + dry) NHx deposition. This implies that the ecosystems are taking up little to no deposited NHx, which does not seem to be a reasonable result during the growing season. This casts doubt that any improvements in model performance is for the "right reasons" and on the value of the source apportionment results. I think that the authors should investigate and discuss the net total reduced nitrogen deposition, and if they cannot justify the high emission/reemission rates of ammonia, then I question the value of the final source attribution results.*

While the simplification in our scheme means that the soil and canopy pools of NH4+ are "infinite", which is not realistic, we can justify it for the following reasons. First, we note that Zhu et al (2015) use this method for the canopy pool of NH4+ in GEOS-Chem (used empirical average values for constant stomatal NH4+, which essentially makes for an infinite canopy pool). And while they more realistically model the soil pool, they required a 3-month spin up to get the soil pool stable. This means that the soil pool is sufficiently large that once stable it may be considered an infinite source, especially over shorter time scales, such as those we have used in our study; our assumption that the pool won't get depleted is reasonable. Second, we have replotted the modelled deposition, combining the dry NH3 deposition + the wet NH4+ deposition to get a total NHx deposition. For the base, bidi, and fire+bidi, the results of the total deposition are shown below:

[Figure]

**Figure R.1:** Total deposited NHx (dry NH3 + wet NH4+) in (a) base, (b) bidi, and (c) fire+bidi

Here we can better see that the ecosystems are in fact taking up deposited NHx over most of the domain over the time period we have simulated (anywhere that's blue is net deposition, anywhere that's red is net upward flux) – which was not easy to see when the deposition maps were presented separately (e.g., Figures 12 & 14 for dry and wet dep, respectively, in the original manuscript). In the revised manuscript, we will present and discuss Figure R.1 instead of showing the two separately since the total deposited NHx is the more important and relevant value. The average NHx flux values across the domain are:

| NET FLUX (moles/m2/day) | Base | Bidi | Fire+bidi |
| --- | --- | --- | --- |
| Mean | -3.025E-5 | -1.811E-5 | -3.765E-5 |
| Median | -2.061E-5 | -1.299E-5 | -2.843E-5 |

From these numbers you can see that in fact, the mean net flux of NHx across the domain from each simulation is similar and is net *downward* (negative). In fact, the fire+bidi has the largest mean net flux downward. Thus, our bidi scheme – even with a soil pool that can't be depleted – does not cause unrealistic net upward flux. In fact, Figure R.1c, shows that there is net deposition where NHx atmospheric concentrations are highest, but that parts of the domain where NHx atmospheric concentrations are low have a net upward flux.

Addressing those "red" areas in Figure R.1; While the red areas in Figure R.1 have net upward flux during our study's time period, it is important to note that our study occurred during August and September, which are very warm months, and the compensation point increases exponentially with temperature (Figure R.2 showing an example for one of the dominant land use categories in the northern part of the domain).

[Figure]

**Figure R.2:** Compensation point (Cg) relationship to temperature; Cg for evergreen needle leaf LUC shown as example.

The higher the compensation point value is, the greater the likelihood there will be upward flux, and the lower it is, the greater the likelihood there will be deposition. Therefore, during the rest of the year (e.g., the preceding winter and spring), the compensation point is much lower, greatly increasing the likelihood of net deposition, even in the areas shown as net upward flux in Figure R.1. This seasonal effect of winter filling the reservoir should be more pronounced with increasing latitude. While we did not run our bidirectional flux simulation for the whole year, a standard (deposition only) GEM-MACH run for the whole year of 2013 yielded a cumulative NHx (wet NH4 + dry NH3) deposition that was greater than our upward flux for Aug/Sept. This means that we can expect the soil pool to be replenished during cooler times of the year, rather than depleted. We therefore feel that our modelling assumptions in this study – especially

given that we modelled a short time period in the summer – are justified. The above discussion will be added to the revised manuscript.

---

## Author Comment (AC2) · 20 Dec 2017

[CW] Thank you for your thorough review of our paper. We have taken into account all of your suggestions, and it has greatly improved the revised manuscript. Please see below for item-by-item responses to each comment (our responses start with "[CW]"). Please also find the revised manuscript attached for reference to the line numbers we mention. We will also be uploading the manuscript with changes tracked.

Reviewer #1's comments:

Citations of non-available papers are given (cited as "this issue" but non available in the

special issue). This stands for example for description of measurements (Wentworth et al), deposition of N and S (Makar et al), parameterizations (Akingunola et al), fire emissions (Zhang et al.).

[CW] This is one of the difficulties with having a number of linked papers being submitted at the same time. We have modified the text to only refer to the papers which have been submitted at the time these revisions have been carried out. Some of these citations have been submitted to the special issue (e.g., Makar, Zhang), and should appear on ACPD soon. When they are not (e.g., Akingunola, Wentworth), the references were removed.

NH3 concentrations are really low in the region of study and the reader may wonder why this region is worth studying, in the light of what is written in the introduction, (NH3 may be harmful for air, water quality, or ecosystem and human health). A sentence or 2 on the relevancy of studying regions where concentrations remain low for the moment would be useful.

[CW] While the NH3 concentrations are relatively low (0.6 ppbv), the impact on atmospheric chemistry may be large, via the formation of particle ammonium nitrate and ammonium sulphate, and through acidifying emissions. It has been shown that ecosystems in Northern Alberta and Saskatchewan are sensitive to nitrogen deposition, and that dry deposition of NH3 and NO2 dominate the near-source N deposition, while wet deposition of ammonium ions dominate the long-range transport ammonia budget (Makar et al 2017, this ACPD issue). The amount of gaseous ammonia in the atmosphere is thus crucial for accurate estimation of the deposition of nitrogen in the region. It has also been shown the emissions of other pollutants from the AOSR are comparable to that of a city (e.g., Liggio et al, Nature, 534, 91-94, 2016), thus, even where NH3 concentrations are relatively low, it is important to understand if the OS facilities are causing critical levels of NH3, and if not, if any other kinds of sources (e.g., fires, re-emissions) are. This justification has been added to the revised manuscript (lines 63-69).
Some citations are given for satellite measurements of NH3 concentrations at a large scale, but nothing is said about orders of magnitude. Do the model results of this study match with previous satellite measurements? More values of ambient concentrations should be cited.

[CW] The model results of this study match the magnitude of NH3 given by TES in the Shephard et al (2015) paper. The model values in southern part of our domain also match the high end of values reported by NH3 reports (e.g., ref AMoN report for NH3 levels in agricultural and other regions). We have added this additional discussion to the revised manuscript (lines 80-85, 484-486). Clarisse et al (2009) and Van Damme et al (2014) report on IASI NH3 global measurements – however these are total columns, thus harder to compare to the CrIS profiles and surface concentrations in ppbv. Zhu et al (2013) also use TES, but their measurements focus on the U.S., thus would be redundant and less relevant study than the Shephard et al (2015) study mentioned above. Similarly Beer et al (2008) also use TES and focus on the U.S. and China. Low NH3 observations were also shown in the Supplemental Information of Kharol et al (2017, GRL), but these were for Alaska and Yukon, and from the CrIS satellite, so would be redundant to mention.

This is also true for bidirectional exchanges: some papers are cited, but only as a list of papers, and no quantified values are given to be compared with what is found in this study. The papers should be cited with precise examples of measured fluxes is the same area or in regions with the same type of ecosystems.

[CW] Zhu et al (2015) found with GEOS-Chem that the re-emission of NH3 added around 1 ppbv to NH3 concentrations globally in the month of July in North America (but decreased concentrations during the cooler months), and that bidirectional flux did not increase NH4 wet deposition. Wichink Kruit et al (2010) found something more similar to us in their 2007 European study, which was a decrease in NH3 deposition, but an increase in wet NH4 deposition. We have added this information, along with some quantitative comparisons to other sources in the literature (e.g., Kharol et al

(2017); Behera et al (2013), into our discussion in Section 5 (lines 603-607, 631-362, and 644-652).

A meteorological description of the site would be useful all along the study. Indeed, emissions have their impact on NH3 concentrations, but wind speed, humidity, temperature have also a significant impact on exchange fluxes. This part of dynamical interpretation on deposition fluxes is missing.

[CW] The meteorological description and discussion of the region and its impact on bidirectional flux is now added to the revised manuscript, in Sections 2.2 (bidi description), 3.1 (surface site description), Section 5.2 (effect on deposition), and the conclusions.

A discussion about why the addition of bidirectional flux is so important in improving the model is missing in terms of processes. The discussion is only about ppb and %, and not about processes.

[CW] Section 2.2 described the bidirectional flux process, and where the temperature dependency comes in (e.g., equation 3), but a lot of the dynamics/meteorology was hidden in the resistance terms (Ri). We have added additional discussion about how meteorology plays a role to this section (lines 218-230) as well as when discussing deposition in the results section (Section 5.2, lines 639-641). We also added a better explanation of the process at the beginning of Section 2.2, lines 174-179.

2- Does the paper present novel concepts, ideas, tools, or data?: not really, bidirectional exchanges of NH3 is already known to be important, tools and data have already been used in Shepard et al (2015)

[CW] We would argue that our paper presents novel tools (GEM-MACH-Bidi) and data (CrIS observations, and FireWorks emissions): The Shephard et al (2015) study used TES satellite observations and a version of GEM-MACH that did not have bidirectional flux or forest fire emissions. Whereas, in our study, we have used new CrIS satellite

observations and the GEM-MACH-Bidi model, which is a new version of GEM-MACH that has the bidirectional flux scheme (previous GEM-MACH versions had no NH3 re-emission process). GEM-MACH-Bidi is the new tool, which we evaluate with cutting edge NH3 satellite data (from CrIS). We note that ours is the first study to use the CrIS satellite observations of NH3 for model evaluation. Our study is also the first to use the FireWorks forest fire emissions at such high spatial resolution. We have updated the abstract, introduction, and conclusion to emphasize these novel features of our study so that they are clearer for the reader.

11- Is the language fluent and precise? Not always

[CW] The language was made more precise in the revised manuscript, where the reviewer mentioned problems in the detailed comments.

13- Should any parts of the paper (text, formulae, figures, tables) be clarified, reduced, combined, or eliminated? Yes, mentioned in specific comments. 14- Are the number and quality of references appropriate? Not always

[CW] When specific comments were made relating to these issues, they were addressed in the responses below.

Specific comments Line 42: As you write that NH3 is a contaminant, precise in what order of concentration it has negative effects.

[CW] While NH3 is poisonous if inhaled in great quantities, these are much greater than found in the atmosphere. For example, there is an Alberta Ambient Air Quality Objective for NH3, which is 2000 ppbv 1-hour average; the basis for this is odour http://aep.alberta.ca/air/legislation/ambient-air-quality-objectives/documents/AAQO-Summary-Jun29-2017.pdf , but this is not relevant to the outdoor atmospheric conditions we are talking about in our study. The reason NH3 was listed as a Criteria Air Contaminant is because of its secondary effects as a PM precursor (Environment Canada, 2001), as discussed in this paragraph of the origi-

nal manuscript. Therefore, one cannot say what ambient concentration of NH3 would cause negative health effects because it is complicated by the atmospheric conditions (meteorological and concentrations of other chemical species) for PM formation. Thus, there is no federal Canadian Ambient Air Quality Standard for NH3, however, there is for PM2.5 (in Canada, the PM2.5 guideline is 28 ug/m3, 24 hour average). Another negative affect of NH3 is its contribution to nitrogen deposition (discussed in the Makar et al companion paper) – wet ammonium ion deposition dominates N deposition in the regions hundreds of km downwind from the anthropogenic sources, for example. The amount of nitrogen deposition which may cause an exceedance of critical loads is highly dependent on the local ecosystem characteristics, and varies by over 3 orders of magnitude in the region examined here. Thus, no particular NH3 concentration can be cited to cause an exceedance in critical loads.

Precise somewhere in your introduction that despite negative effects of high concentrations, low concentration regions are also worth studying.

[CW] It is still important to study this region because the modelled background NH3 must be correct in order to understand the relative impacts of the oil sands operations, and because even a small amount may cause a critical load exceedance for deposition to sensitive terrestrial and wetland ecosystems. This explanation was added to the introduction, lines 63-69.

Line 44: Modeling provides: : :: this could be true if inventories are correct and with fine resolution, which is hardly feasible in most models. Remove this first part of the sentence.

[CW] Done.

Line 55: reformulate your sentence because reading "the AOSR is a large source of air" is a bit weird.

[CW] Done.

[Figure]

Line 58: why are NH3 concentrations so low, despite local pollution?

[CW] The anthropogenic emissions of the AOSR take place in a relatively small region compared to all of "northern Alberta and Saskatchewan", which is where we've said that NH3 concentrations are low. We were basically saying that the area surrounding the AOSR has very few sources of NH3, and as a result, background concentrations of NH3 in that area are low (b/c of little population and lack of agriculture, compared to the southern part of the province. The low background concentration puts into context the ~0.5 ppbv model bias that existed before the addition of the missing sources. We've reformulated those sentences in the revised manuscript (lines 63-65) to better get that point across.

Line 61: give mean concentrations of cited agricultural areas.

[CW] In our reformulation, this reference is removed.

Line 66: give values of the fraction of deposited NH3 compared to NO2 and HNO3.

[CW] Since our references were based on atmospheric concentrations rather than deposition measurements, we have revised the wording here, and given ratios of NH3 to other N gases in the air. See lines 72-76 in the revised manuscript.

Line 114: precise what species are used from the inventories.

[CW] List of 25 species was added to the manuscript (lines 128-131).

Line 121: A word about the importance of carefully design stack parameters would be useful to understand why this part is so important for your study.

[CW] On line 128 of the original manuscript we mentioned that ill-designed stack parameters resulted in "erroneous short term plume events", causing "NH3 levels up to 2 orders of magnitude higher than ground observations", which should be explanation enough for the importance of stack parameters. However, we have added a note in the revised manuscript (line 148) that this stack is in the AOSR, thus it is important to get

right for our study.

Line 167: give reference values of realistic NH3 concentrations consistent with findings from the literature, and explain why you use low end values.

[CW] We referenced annual AMoN values for "realistic NH3 concentrations" (line 209). The explanation of why the low end values were chosen was already given – to get realistic NH3 concentrations compared to observations.

Line 191: what do you mean by "major point emissions?"

[CW] An explanation of area and major point emissions has been added to the Emissions Section (Sec 2.1), lines 137-140.

Line 220: what is ECCC?

[CW] Acronym is now defined upon first use.

Line 259: you mention +/- 15% of uncertainty for measurements, with a mean measured value of 0.63 ppb (line 59). The model/measurements bias is 0.3-0.85 for bidi, what is the real impact on modeled concentration? What is the range of possible concentration? Is the measured concentration included in this range?

[CW] For the AMS-13 model evaluation (a single site heavily influenced by local anthropogenic sources), a 15% error on the measured 0.63 ppbv average is only 0.0945 ppbv, giving the range of 0.536 ppbv to 0.725 ppbv for the average measured concentrations there. The median model bias of the base case is -0.35 ppbv, meaning that the model reports only 0.28 ppbv NH3 in that area – well below the bottom of the measurement range. The bidi model had a +0.3ppbv bias, meaning that the model reports 0.96 ppbv in that area, which is well above the range in the measurement average. Taking the standard deviation of the model biases at that specific location into account, and they all overlap with the measurements (e.g., the vertical range in the whiskers in Figure 5). However, we see in Figure 6, that the real improvements come in the form of better correlation and slopes with the bidi and fire+bidi models. We have not added

any new text, as the existing text covers all of this.

Line 299: Figure 3 should not be placed here. It is only used later in the text, and should be included before figure 9. Furthermore, if placed here, it is not understandable why these three dates are chosen (not explained in the caption).

[CW] We have removed the reference to Figure 3 early on, and moved the figure to the appropriate place in the results/discussion.

Line 324: You mention fig 5 and then you talk about fig 4. Place fig 5 after fig 4 and give a description on interpretation of it.

[CW] That sentence about Fig 5 is removed, and figure 5 discussion is now appropriately placed.

Line 326: what are the background times?

[CW] Changed to "when NH3 concentrations are relatively low (< 0.5 ppbv in the base model)".

In figure 4, NH3 is in ppb, but NH4+, NO3- and SO42- are not in ppb. Please correct the caption

[CW] Done.

Line 335: what is the increase of concentration with the influence of a local plume? 0.5 ppb seems to be very very low, and more included in the measurement noise than in a local pollution signal.

[CW] The 0.5 ppbv reference is in regard to the base model (line 334 of original manuscript) – not in the measurements. The base model has very very low background concentrations (which we are correcting in this work), thus any time the base model goes above 0.5 ppbv, we can assume a nearby source, such as a plume. Note that even the measurements did not exceed 3 ppbv, and had a mean of 0.6 ppbv during this time period. Thus 0.5 ppbv is not low in relation to that. Discussion of these

concentrations can be found in lines 387-392 of the revised manuscript.

Line 337: 0.08 ppb is less than the 15% measurement uncertainty. Is it really significant?

[CW] No, you are correct, it isn't. Therefore, we have removed the discussion about removing the plume influence to see "better" results, since they are not significantly better.

Line 339: R=0.1 and 0.4. Are these coefficients significant? Could you give a significance (p-value) of your correlation calculation everywhere it is necessary?

[CW] We have done a paired t-test on the AMS13 surface NH3 data, and found that none of the three model simulations can be considered statistically indistinguishable from the measurements at the hourly time scale (t>1, p<0.05), although the weekly-averaged bidi simulation comes close, with t=1.9, p=0.15 (for model to be considered the same as the measurements, t should be <1, and p should be >0.05). These statistics are now included in Table 2 and discussed in the revised manuscript. Since the AMS13 results are just for a single gridpoint at the surface (where local variability due to point sources makes getting a good match closer to sources difficult), we have also added statistics for the aircraft and satellite results as well. The satellite results cover a much larger domain (all of Alberta and Saskatchewan, and throughout the troposphere). For the particulate species, none of the correlations are significant either. Most air quality models do not model PM species well, and this is an area of on-going study.

Line 340: Fig 6a: what is the unit of NH3 concentration?

[CW] Fig 6a has units of ppbv in the axis labels, and the slope of the model/measurement line is unitless. Nothing changed.

Line 368: SO42- is influenced by anthropogenic emissions, why not by fire emissions?

[CW] Added "and fire emissions" here.

Line 375: Why did you choose this precise flight? By the way, it would be useful to give some average values of meteorological conditions when describing the area of study (mean temperature, humidity, rainfall, wind speed, etc: : : all parameters that have a possible influence on NH3 concentration)

[CW] This flight was chosen as an example because this flight sampled mainly background NH3 concentrations (rather than facility plumes), and it is the modelled background NH3 that this study aims to improve (lines 448-450 in revised manuscript). Meteorological conditions for the AOSR were added to Section 3.1, lines 317-329.

Line 376: can you explain why fire+bidi does not improve the results compared to bidi?

[CW] In Figure 7c and d, there is very little difference in concentrations because the flight did not pass through a fire plume. We have added this explanation to the revised manuscript (lines 456-457).

Line 404: Figure 3 is used in this paragraph. It should appear in the text at this time, and not before.

[CW] Done: Fig. 3 of the original manuscript, is now Fig. 8 in the revised manuscript.

Line 430: specify at the lowest level.

[CW] Done.

Line 446: remove "that" before the fire+bidi model

[CW] Done.

Line 451: You suppose that the conversion of NH3 to NH4+ is underestimated: did you have a look at the NH4+ pool in that case?

[CW] Unfortunately, there were no NH4+ measurements in that region to compare to.

Line 453 to 459: this explanation is very confused. Please rephrase.

[CW] It has been rephrased and put into point form, making clear which part of the

discussion is for which explanation (a, b, c) of the bias (lines 549-569).

Line 507: this sentence is not useful. Obviously if bidirectional parameterization is used it will balance deposition with emission and not increase deposition fluxes.

[CW] We agree with the reviewer that using a bidirectional scheme can only decrease net deposition for NH3 compared to a traditional unidirectional deposition scheme if all the other model components/parameters remain the same, e.g., the Zhang et al. (2010) bidirectional scheme versus the unidirectional scheme of Zhang et al. (2003), because the former was built on the later. However, the original unidirectional scheme used in our model was not exactly the same as in Zhang et al. (2003), but a hybrid of Wesely (1989) (for stomatal uptake) and Zhang et al. (2003) (for non-stomatal uptake). Thus, we cannot completely exclude the possibility that the new bidirectional scheme might produce higher deposition under certain circumstances without a model validation.

Line 517 and elsewhere: the unit for the flux is not appropriate. Please homogenize throughout the paragraph and use preferably ngN.m-2.s-1.

[CW] Units of moles/m2/day are now consistently used throughout.

Lines 584 and 587 use another unit which is not a flux unit. In this paragraph needs bibliography values need to be included for equivalent ecosystem or region.

[CW] Do you mean lines 582-583, which talked about % contributions to atmospheric concentrations? No numerical values were written on lines 584 and 587. However, we had included references to reported deposition and discussed them in Section 5.2, lines 524-531 of the original manuscript, using appropriate units. The revised manuscript has flux in moles/m2/day throughout.

Figure 12 and 13 are redundant.

[CW] Figure 12-14 were removed, and replaced with a new Figure 12 that is the total NHx deposition.

Figure 13 is not necessary in my opinion.

[CW] Figure 12-14 were removed, and replaced with a new Figure 12 that is the total NHx deposition.

Line 539: again this sentence is not useful. Obviously wet deposition is only deposition. Again flux units are not correct and should be homogenized.

[CW] Any mention of ïA■moles/m2 should have been moles/m2/day. This was corrected in the revised manuscript. Also, most of the discussion is now about total NHx deposition, rather than of wet and dry deposition discussed separately.

Line 550: how do wet deposition fluxes compare with literature?

[CW] It has been difficult to find relevant wet deposition fluxes in the literature, as many of those studies report on ammonium concentrations in rainwater without giving the precipitation flux. Or they report on locations that are not appropriate to compare to Alberta/Saskatchewan (e..g, Murano et al, 1998 found average values of 1E-4 moles/m2/day in Japan). We have however, added a couple of references, including a technical report in the United States, which reported about 2E-5 moles/m2/day average in the U.S. (lines 668-671). Our results are in between the values reported in those two studies.

Line 570-571: this sentence has already been written above. The conclusion should mention the possible influence of meteorological conditions on NH3 concentration, as well as in the text.

[CW] The fact that the NH3 emission factors need to be revisited for further model improvements is an important conclusion of our study, and that is why we have highlighted it again in our conclusion. The meteorological discussion of the region and its impact on bidirectional flux is now added to the revised manuscript, in Sections 2.2 (bidi description), 3.1 (surface site description), Section 5.2 (effect on deposition), and the conclusions.

[Figure]

Line 574: "miniscule": please quantify.

[CW] 0.02 ug/m3 for each. This was added to the sentence (lines 698-696).

Technical comments Line 16: remove , aftertime period.

[CW] Done.

Line 108: write covers instead of covering, twice in the line.

[CW] Done.

Line 109: remove "And" at the end of the line.

[CW] Done.

Line 117: remove ( before Zhang and put ( before 2017.

[CW] Done.

Line 163: replace "in" by "from" before Wen et al.

[CW] Done.

Line 228: Time period "from"

[CW] Done.

Line 296: a verb is missing in the sentence.

[CW] The verb was "compute", but it should have been part of the list. We have fixed the sentence.

Line 321: include "to" after compared.

[CW] Done.

Line 352: remove italics for "should", same line 488 for "total" and line 507 for "more"

[CW] Done.

[Figure]

Line 454: a ) is missing after 10c

[CW] Done.

Line 493: problem with the sentence, please rephrase

[CW] Done.

Line 507 and 509: remove , after 11.

[CW] Done.

Line 526: replace x by times

[CW] Done.

Please also note the supplement to this comment:
https://www.atmos-chem-phys-discuss.net/acp-2017-627/acp-2017-627-AC2-
supplement.pdf
* * *
[Figure]

**Supplement:**

[revised manuscript text omitted]
 (3 Sept, 1 Sept and 12 Aug, 2013) that we use for the case studies, and surface $NH_3$ concentrations over that region as well as sample Aqua MODIS true colour composite maps for those days are shown (Fig. 10). The four boxed regions on those maps indicate where model-measurement pairs were sampled for this study. The cyan and black boxes in Fig. 10a
510    and b are the regions where we sample clear-sky, no-fire conditions on 3 and 1 September 2013, respectively. The magenta box in Fig. 10c is the region where we isolated our fire case study on 12

August 2013. The blue box is the region we discussed above, which we analysed for the full time period simulated (12 Aug - 7 Sep 2013, Fig. 9top).

**4.3.1 Case study 1: clear-sky days with little fire influence - evaluating bidi**

515 In order to evaluate the bidirectional flux component separately from the fire component, we selected September 1st (southern, agricultural region - black box in Fig. 10b), and 3rd (northern, boreal forest and AOSR region - cyan box in Fig. 10a), where the MODIS map (EOSDIS NASA World view map, worldview.earthdata.nasa.gov) shows very little hot spots from fires, and that the conditions were relatively cloud and smoke free (which yield the most CrIS observations). See Table 1 for the

520 latitude and longitude ranges. Figure 10 also shows the surface $NH_3$ concentrations as observed by CrIS on each of those days. Figure 11a shows that in the north, the bidi model improves the bias from -0.84 ppbv to -0.07 ppbv in the lowest vertical level, and smaller, but still significant, improvements to the bias at the other levels. The fire+bidi model has a nearly identical impact as the bidi model, which is expected in a fire-free zone. Therefore, the GEM-MACH-Bidi model performs very well in

525 northern Alberta and Saskatchewan where there is mainly boreal forest, and background-level $NH_3$ concentrations. This also implies that the LUC assignment discussed in Section 4.1 may only apply to a small region around the AOSR, and not to the overall large region we've defined here.

In the southern region (Fig. 11b), the addition of bidirectional flux moves the bias from near-zero to +1.02 ppbv in the lowest level. In this case, the base model with no bidirectional flux appears to be

530 the most accurate model in areas dominated by agricultural sources. There are two possible explanations: a) agricultural emissions are too high in the base model, and the addition of the bidirectional flux leads to an overestimation of the $NH_3$ amounts, or b) re-emissions from bidirectional flux from crops are not significant. The literature (Bash et al., 2010; Massad et al., 2010; Zhang et al., 2010; Zhu et al., 2015) indicate that crops do indeed re-emit $NH_3$, therefore, (a) is the more likely expla-

535 nation. The agriculture $NH_3$ emission inventory we used was created by the NAESI (National Agri-Environmental Standards Initiative) project (Bittman et al., 2008; Ayres et al., 2009; Makar et al., 2009) have about 30-200% uncertainty associated with them (Bouwman et al., 1997; Asman et al., 1998). Therefore, with improved national $NH_3$ emission inventories, the GEM-MACH-Bidi should improve model results across the domain.

**4.3.2 Case study 2: a clear day with significant fire influence - evaluating fires**

In order to evaluate the fire component separately from the bidirectional flux, we selected August 12th (a northern region with little-to-no agricultural contributions) where the MODIS map shows numerous hot spots from fires and smokey conditions (Fig. 10c, magenta box). The base and bidi models underestimate $NH_3$ concentrations (Fig. 11c ) by -6.22 and -5.84 ppbv, respectively (in the

545 lowest vertical layer), but the fire+bidi model overestimates $NH_3$ by +4.06 ppbv. The fire+bidi version of the model still has the lowest bias of the three simulations, however, either (a) the fire+bidi

model does not distribute the fire emissions properly in the vertical, (b) the fire emissions of $NH_3$ are too high, and/or (c) the model is not properly representing $NO_2$ and $SO_2$ in the fire, and so the conversion of $NH_3$ to $NH_4^+$ is underestimated. It is potentially a combination of all three explanations,

550     and we further elaborate below.

– For explanation (a), both fire plume rise and fire emission factors are on-going areas of study. In the model the fire emissions are distributed evenly throughout the boundary layer (the first 3-4 layers in Fig. 11c), however, Shinozuka et al. (2011) suggest that sometimes the fire plumes are distributed normally in a thin layer aloft. However, should that be the case for the

555     real-life fires in this case study, the model bias would be negative at at least one of the levels in Fig. 11c, which it is not. Figure 11c shows that the positive bias extends throughout the first three vertical layers, and in the top two vertical layers, the bias does not move further negative (as would happen in the fire plume were actually at those altitudes in real life).

– (a) Our bias would be very high at low levels if the real fire plumes were actually above 4

[revised manuscript text omitted]

---

## Author Comment (AC3) · 20 Dec 2017

[CW] Thank you for your thorough review of our paper. We have taken into account all of your suggestions, and it has greatly improved the manuscript. Please see below for item-by-item responses to each comment (responses start with "[CW]"). Please see our response to the other reviewer for the attached supplement, which is the revised paper, to which the line #s refer to.

Serious concern: In the authors' implementation of the bidirectional NH3 flux, they assumed that there was an infinite soil pool of NH4+. This is an unreasonable assumption that is recognized and discussed by the authors. However, due to this or

other assumptions in the implementation of the NH3 bidirectional flux mechanism, the NH3 emission/reemission flux is similar to or greater than the total (wet + dry) NHx deposition. This implies that the ecosystems are taking up little to no deposited NHx, which does not seem to be a reasonable result during the growing season. This casts doubt that any improvements in model performance is for the "right reasons" and on the value of the source apportionment results. I think that the authors should investigate and discuss the net total reduced nitrogen deposition, and if they cannot justify the high emission/reemission rates of ammonia, then I question the value of the final source attribution results.

[CW] As you say, we recognized that this simplification (using empirical average emission potentials) means that the soil and canopy pools of NH4+ are "infinite", which is not realistic. First, we note that Zhu et al (2015) use this method for their canopy pool of NH4+ in GEOS-Chem (used empirical average stomata emission potentials, which essentially makes for an infinite canopy pool). While they more realistically model the soil pool, they required a 3-month spin up to get the soil pool stable. This means that the soil pool is very large, and that over the shorter time scales we use in our study, assuming that the pool won't get depleted is a valid assumption. This is further supported by Wentworth et al (2014, Biogeosciences), who calculated the approximate relative abundances of NHx in the boundary layer versus NH4+ in the soil pool to assess whether surface-to-air fluxes were sustainable. They found that soil NH4+ » boundary layer NHx (by over two orders of magnitude), further supporting the assumption in our bidirectional flux scheme. In addition, the turnover time for soil NH4+ is on the order of 1 day, and the majority of soil NH4+ comes from org-N decomposition (Booth et al., 2005, Ecol. Monogr.), hence it is unlikely that NH3 bi-directional fluxes would significantly deplete/enhance soil NH4+ pools over shorter time scales such as the month simulated here. Second, we have replotted the modelled deposition, combining the dry NH3 deposition + the wet NH4+ deposition to get a total NHx deposition. For the base, bidi, and fire+bidi, the results of the total deposition are shown below:

(see attached Fig_R1) Figure R.1: Total deposited NHx (dry NH3 + wet NH4+) in (a) base, (b) bidi, and (c) fire+bidi. Red regions indicate net NHx emissions; and blue regions indicate net deposition.

Here we can better see that the ecosystems are in fact taking up deposited NHx over most of the domain (anywhere that's blue is net deposition, anywhere that's red is net upward flux) – which was not easy to see when the deposition maps were presented separately (e.g., Figures 12 & 14 for dry and wet dep, respectively, in the original manuscript). In the revised manuscript, we will present and discuss Figure R.1 as the new Figure 12, instead of showing the two separately since the total deposited NHx is the more important and relevant value. The average NHx flux values across the domain are: NET FLUX (mol/m2/day) Base Bidi Fire+bidi Mean -3.025E-5 -1.811E-5 -3.765E-5 Median -2.061E-5 -1.299E-5 -2.843E-5 From these numbers you can see that in fact, the mean net flux of NHx across the domain from each simulation is similar and is net downward (negative). In fact, the fire+bidi has the largest mean net flux downward. Thus, our bidi scheme – even with a soil pool that can't be depleted – does not cause unrealistic net upward flux. In fact, Figure R.1c, shows that there is net deposition where NHx atmospheric concentrations are highest, but in parts of the domain where NHx atmospheric concentrations are low there is a net upward flux.

Addressing those "red" areas which are still visible in Figure R.1b and c; While the red areas in Figure R.1 have net upward flux during our study's time period, it is important to note that our study occurred during August and September, which are very warm months (discussion of meteorological conditions in the region was added to the revised manuscript), and the compensation point increases exponentially with temperature (Figure R.2 showing an example for one of the land use categories in the northern part of the domain).

(see attached Fig_R2) Figure R.2: Compensation point (Cg) relationship to temperature; Cg for evergreen needleleaf LUC shown as example.

The higher the compensation point, the more likely there will be upward flux, and the lower it is, the more likely there will be deposition. Therefore, during the colder part of the year (e.g., the preceding winter and spring), the compensation point is much lower than during our study, increasing the likelihood of net deposition, even for the regions shown as emitters in the summer in northern Alberta/Saskatchewan in Figure R.1. While we did not run our bidirectional flux simulation for the whole year, a standard (non-bi-di) GEM-MACH run for a full year, yielded a cumulative NHx (wet NH4 + dry NH3) deposition that was greater than our upward flux for Aug/Sept. This means that we can expect the soil pool to be replenished during cooler times of the year, rather than depleted. Thus, our modelling assumptions in this study – especially given that we modelled a short time period in the summer – are justified. This discussion, figure, and table have been added to the manuscript in Sections 2.2 and 5.2 of the revised manuscript.

The authors pursued the incorporation of ammonia bidirectional flux and wildfire emissions into the model due to significant underestimations of ammonia concentrations in a previous modeling exercise. While reasonable, they do not discuss potential issues with other modeling inputs and processes, including the underestimation of emissions from other sectors, e.g., agricultural regions and NH3 slip in fossil fuel combustion systems, as well as potentially overestimating NH4 wet deposition. Early in the manuscript it would be good to discuss why these other factors are not likely significant contributors to the initial model underestimation. This could include evaluation of the model NH4 wet deposition simulation against measured wet deposition or through fall data. If NHx wet deposition is also underestimated, then this would certainly point toward biases in the dry deposition rates and/or emissions.

[CW] We presented evidence in our study that agricultural emissions of NH3 are likely overestimated. We know GEM-MACH's NH4 deposition is not overestimated because of work in Makar et al (2017, in this special issue of ACPD), where they showed a small underestimation of NHx deposition in the base (non-bi-di) GEM-MACH model (model

to observation slope of wet deposited nitrogen of 0.89, R2 = 0.76 ). We've added this point to introduction, lines 85-89, in the revised manuscript.

Near the end of the manuscript, the authors do show that the base-case model simulation performed well near agricultural activity and that it underestimated NH3 when wildfire emissions impacted the area. This information supports the authors' premises, and I suggest that these results be discussed before the model comparison to the surface and aircraft measurements.

[CW] The following text was added to the introduction (lines 85-89 in the revised manuscript) to motivate the two changes we made to the GEM-MACH model in this study: Having too much modelled NHx deposition is a cause that was ruled out when Makar et al (2017) showed that GEM-MACH actually underestimates NHx deposition. Underestimating anthropogenic and agricultural emissions in southern Alberta and Saskatchewan was also ruled out as a cause because the GEM-MACH model performs well in southern Canada and the U.S when compared to the U.S. Ambient Ammonia Monitoring Network (AMoN). NH3 sources known to be missing from the GEM-MACH model were forest fire emissions and re-emission of deposited NH3 from soils and plants (the latter referred to as bidirectional flux, hereafter), which would have the greatest impact in background areas, such as northern Alberta and Saskatchewan. Therefore, these two sources were added to an updated version of GEM-MACH...

Last, the oil sands region is an area of intense energy development, and some discussion of the ammonia emission from this activity and its uncertainty is warranted.

[CW] In another companion paper being submitted to the special issue the emissions are discussed in detail (Zhang et al). In the province of Alberta, the reported oil sands emissions represent 1% of the province's total anthropogenic NH3 emissions. The oil sands have two different emissions inventories: the National Pollutant Release Inventory (NPRI) annual inventory, and Continuous Emissions Monitoring (CEMS) hourly emissions data. The CEMS emissions have relatively

low uncertainties because they are based on measurements in the stacks. However, only some of the facilities measure NH3 emissions. Those that do would base their reported NPRI emissions on those CEMS measurements. Those that don't have higher uncertainty on the NH3 emissions they report. For example, the Syncrude facility has CEMS-based NH3 emissions in the NPRI inventory, so it should have relatively high quality (see http://www.ec.gc.ca/inrp-npri/donnees-data/index.cfm?do=substance_details&lang=En&opt_npri_id=0000002274&opt_cas_number=NA%20-%2016&opt_report_year=2013 for NH3 emissions for this facility and http://www.ec.gc.ca/inrp-npri/donnees-data/index.cfm?do=substance_details&lang=En&opt_npri_id=0000002274&opt_cas_number=NA%20-%2016&opt_report_year=2013 for the Basis of Estimate Codes. "M1" means "Continuous Emission Monitoring - In use from 2003 and onward"). However, because we don't have hourly CEMS NH3 emissions for 2013, it is hard to tell the difference between CEMS and NPRI values. Some of this discussion has been added to the manuscript, lines (142-147).

Specific comments In the abstract and introduction it is noted that the Alberta oil sands region has relatively low ammonia concentrations. Please put this into some context. These concentrations are not low compared to many rural western North American sites.

[CW] The low NH3 concentrations are mainly across northern Alberta/Saskatchewan, but not necessarily within 10 km of the AOSR industries. We have modified the text to reflect that distinction (lines 63-71), however, 0.6-1.2 ppbv range that we find in the AMS13 measurements are on the low end of the NH3 2012 annual averages reported in this AMoN data summary: http://nadp.sws.uiuc.edu/amon/ . We have added some reported NH3 concentrations across different areas to the revised manuscript (measured via the AMoN network) in the introduction and Section 4.1 (lines 387-392).

Also, can anything be said about the estimated deposition rates in these regions compared to the reactive nitrogen critical loads? If the deposition rates are near or above

the critical loads, then this work could have important policy implications.

[CW] The issue of acidic exceedances of critical loads of sulphur and nitrogen is the focus of the Makar et al study that has been submitted to the oil sands special issue of ACPD, currently awaiting assignment and initial recommendations from reviewers. The modelling carried out there was similar to our base case, but for an extended period of one year (a more relavant time scale for deposition to ecosystems). There, it was shown that anthropogenic sources in the region create sufficient sulphur deposition to exceed aquatic ecosystem critical loads over a large region; nitrogen deposition was not needed to result in exceedances. In that sense, the additional policy implications of nitrogen deposition may be moot. However, the exceedances were higher when N and S were considered together, but the key point with reference to the bi-directional fluxes is that sulphur alone was already sufficient for exceedances. Nevertheless, we are interested in following up the potential for bi-directional fluxes to influence exceedances, in future work. With regards to nutrient N critical loads (i.e., eutrophication critical loads), to our knowledge, there have not been any N-critical loads developed specifically for the oil sands region.

Specific comments Lines 173-175: "the bidirectional flux acts effectively as an additional source of NH3 gas, releasing stored NH3 until and unless the ambient concentration rises to the compensation point concentration." It would be good to discuss the origin of the NH3 in these emissions. That is, is the NH3 originating from the natural processes of the ecosystem or from previously deposited NH3 or a combination of both? Presumably, it is from both. This also has implications when discussing natural versus anthropogenic NH3. The authors assume that all NH3 bidirectional flux emissions are natural; however, if the deposited NH3 originating from anthropogenic sources was reemitted, then this NH3 would have anthropogenic origins. Consequently, not all of the reemitted ammonia due to the bidirectional flux processes is necessarily natural.

[CW] Since the re-emissions are from soils and plants, we have called them natural in the original manuscript, however, you are correct that the sources of NHx available

for re-emissions are from increased deposition because of anthropogenic sources, as well as from natural N2-fixation, organic decomposition, and microbial action. Vile et al (2014, Biogeochemistry) found that in boreal bogs, 90-95% of the NHx pool is from these natural processes, but that's not necessarily true for other land-types. So it's correct to say that the re-emissions are both natural and anthropogenic in origin. Similarly, forest fires provide another source of NHx which may be classified as natural and/or anthropogenic in origin. With the current GEM-MACH-Bidi model, we can't distinguish how much is from each. However, we have revised the text so that the re-emissions are no longer called "natural", but rather "semi-natural" (lines 10-12, and lines 178-179).

Line 186: "it is not desirable for our bidirectional flux scheme to have to rely in advance on another model's output. Therefore, we use this simplified version, and assess whether its results provide a good enough improvement to simulated NH3 for less cost in run time." The authors did not discuss what constitutes a "good enough" model simulation or whether the studied model satisfied this criterion.

[CW] This is a good point, and the phrase "good enough" was removed from the manuscript. The ultimate goal is to have model biases of zero within measurement errors bars, but this is not always possible given the complexities of an air quality model (e.g., there can be errors in modelled meteorology, emissions inventories, emissions spatial and temporal allocations, atmospheric chemistry, etc., etc.). Furthermore, a zero model bias may be achieved, but for the wrong reasons (e.g., knowing certain process/sources are missing, but compensating errors causing the model values to be close to measurements anyway). Thus, a quantitative threshold for "good enough" is not necessarily comprehensive. We do consider the fire+bidi simulation to have satisfied our objective of "improving NH3 predictions" because it has better statistics when compared to a variety of measurements than the base case has (now summarized in Table 2 for all simulations and measurements), and because it contains all of the known missing sources of NH3 for the region. We have revised that text in Section 2.2 (lines

246-247).

In addition, as discussed in the general comments, I question whether the high NH3 emissions resulting from the bidirectional flux mechanism are reasonable or not and suggest further investigation and discussion.

[CW] The simplification in the soil and stomata emission potentials is an appropriate parameterization for reasons stated our response to reviewer#3's first comment (see above).

Line 307: "Figure 4 shows the time series of the concentrations of NH3 and its reaction products, fine-particulate NH4+ and NO3-" This is a confusing sentence.

[CW] Thank you for pointing out the unclear sentence. It has been revised (lines 384-386).

Also please specify if NO3- is only particulate nitrate or if it includes nitric acid.

[CW] It is only particulate NO3-.

Section 4.1: Reproducing the measured hourly ammonia concentrations is very challenging. It would be good to see how the model performs on an aggregated basis as well, e.g., can it reproduce the 24-hour average NH3 values and the average diurnal cycles?

[CW] Figure R.3 below is the timeseries of daily averages, which is clearer and doesn't need to be in log scale. We have replaced Figure 4 of the original manuscript with Fig R.3, and doing so does not much change the discussion that was there previously. We have kept the following two figures the same (with hourly data) in the revised manuscript.

(see attached Fig_R3) Figure R.3: daily average times series at the AMS13 ground site.

Figure R.4 below shows the analysis of day of week, diurnal cycle, etc. that the R

openair package provides – here just for NH3. We see that while the bidi and fire+bidi models now over-predict NH3 concentrations at this single location which is influenced by local anthropogenic sources, the diurnal cycle is better represented in those simulations, compared to the base simulation, which is just spiky at certain hours. The bidi simulation is more similar to the measurements, although the amplitude of the cycle is still underestimated. Similarly the bidi simulation has the closest agreement with the August monthly average (lower-middle panel), and the average of most of the week days (lower-right panel). We have not added Fig. R.4 to the revised manuscript, however, we have added additional text describing these findings (lines 410-414).

(see attached Fig_R4) Figure R.4: time series analysis for NH3 at the AMS13 ground station. Hours are in UTC (subtract 6 to get local time).

It would be good to include estimates of the model error such as the RMS and fractional errors and bias in the model performance statistics.

[CW] RMS model error and fractional errors have been calculated and added into Table 2. For almost all comparison statistics, the fire+bidi simulation has the best results.

Line 333: "(from R=0.2 to 0.4)..." From Figure 6 it looks like the improvement in correlation should be from 0.1 to 0.4.

[CW] Yes, that's been corrected, as were the slopes.

Line 372: "However, we clearly see that for this flight, the bidirectional flux has increased NH3 concentrations, bringing them closer to the measured values." It is not clear from the figure that the model performance has improved, only that the simulated NH3 has increased. It would be good to add performance stats to panels b–c in Figure 7.

[CW] The improvement can be seen by the fact that the bidi and fire + bidi colours now match the colours in the measurement panel (they all use the same colour scale). The median concentrations of each panel are now mentioned in the text (line 455-456).

Lines 425-435: I think this discussion is very important for justifying the modeling refinements and should be moved up front.

[CW] We added to the introduction, lines 85-89.

Technical comments The fonts used in the figures are very small, making text difficult to read. This is particularly the case in Figures 3, 11, 12, 13, and 14 and supplemental material.

[CW] These figures and their fonts were made larger.

Figures 12 and 13 are missing panels.

[CW] To address another reviewer comment, we have remade Figure 12 (which is Fig 13 in the revised manuscript), and eliminated Figures 13 and 14 from the original manuscript.

———————————————————

[Figure]

**Fig. 1.** Total deposited NHx (dry NH3 + wet NH4+) in (a) base, (b) bidi, and (c) fire+bidi. Red regions indicate net NHx emissions; and blue regions indicate net deposition.

**Fig. 2.** Compensation point (Cg) relationship to temperature; Cg for evergreen needleleaf LUC shown as example.

[Figure]

**Surface concentrations at Oil Sands AMS13 site**

[Figure]

**Fig. 3.** daily average times series at the AMS13 ground site.

[Figure]

**Fig. 4.** time series analysis for NH3 at the AMS13 ground station. Hours are in UTC (subtract 6 to get local time).

---

## Author Comment (AC4) · 20 Dec 2017

Dear editor and both reviewers, I have responded to all reviewers comments, which included a revised manuscript (attached to reviewer 1 responses), which I referred to in both of my responses when I stated which line numbers contain the additional or modified text. Here I would also like to attach the track-changes version of the revised manuscript (created via latexdiff), so that you may more easily see the revisions. Note, however, that the line numbers in the track-changes version do *not* correspond to the line numbers in my responses. Thank you.

[Figure]

Please also note the supplement to this comment:
https://www.atmos-chem-phys-discuss.net/acp-2017-627/acp-2017-627-AC4-supplement.pdf
* * *
[Figure]

**Supplement:**

[revised manuscript text omitted]

580    – For explanation (a), both fire plume rise and fire emission factors are on-going areas of study.  In the model the fire emissions are distributed evenly  throughout the boundary layer (the first 3-4 layers  in Fig. 11c), however, Shinozuka et al. (2011) suggest that sometimes the fire plumes are  distributed normally in a thin layer aloft

585    . However, should that be the case

 for the real-life fires in this case study, the model bias would be negative at at least one of the levels in Fig. 11c, which it is not. Figure 11c shows that the positive bias extends throughout the first three vertical layers, and in the top two vertical layers, the bias does not move further negative (as would happen in the fire plume were actually at those altitudes in real life).

- (a) Our bias would be very high at low levels if the real fire plumes were actually above 4 km (above the altitudes we studied), 
[revised manuscript text omitted]

*Acknowledgements.* The project was supported by Environment and Climate Change Canada's Oil Sands Monitoring program (OSM), and the Climate Change and Air Quality Program (CCAP). We would also like to acknowledge the University of Wisconsin-Madison Space Science and Engineering Center Atmosphere SIPS team sponsored under NASA contract NNG15HZ38C for providing us with the CrIS level 1 and 2 input data, in particular Keven Hrpcek and Liam Gumley.

Data availability: The CrIS-FRP-NH3 science data products used in this study can be made available on request (M. W. Shephard, Environment and Climate Change Canada, Toronto, Ontario, Canada). Similarly, the AMS13 observations can be made available on request from Greg Wentworth (AEP), and the model output or code from Cynthia Whaley (ECCC). The aircraft observations are on the ECCC data portal (http://donnees.ec.gc.ca/data/air/monitor/ambient-air-quali

**References**

Akingunola, A., Makar, P. A., Zhang, J., Darlington, A., Li, S. M., Moran, M. D., and Zheng, Q.: Impact of continuous monitoring emissions stack parameters on model simulations using 2.5 km resolution GEM-MACH, Atmos. Chem. Phys. Disc., 2017.

[revised manuscript text omitted]

---

## Editor Decision (ED1)

This has been a long drawn out exercise, but I feel the manuscript is almost ready for publication in ACP, my congratulations. I have a few statements of a general nature, and a quite possibly not exhaustive list of minor issues/corrections:

(1) You have used units of ppbv for NH3. I assume you know that ppbv is *not* a concentration unit (it is dimensionless). It is also not the proper SI unit (should be nmol/mole). I will not bother you with ordering to change "ppbv", but whenever in your text you use "ppbv", you should change the term "concentration" into "mixing ratio", or the more correct term "mole fraction"

(2) As for referencing other papers that (will) belong to the AOSR special issue you should use the actual full ACP reference rather than "this issue". ACP papers are published on line when accepted, and will not be hold back to appear all at once with consecutive numbering, so the term "this issue" does not work. Furthermore, I noticed somewhere a reference to a manuscript in the ACPD phase. That is somewhat acceptable (you refer in that case not to a peer reviewed publication, although the manuscript has been accepted for review, which is something entirely different!) but here even more reference to "this issue of ACDP" is meaningless (manuscripts in ACPD will not appear in a hardcopy version, only online.

(3) I applaud you for the consecutively line numbering of your manuscript. However, you have to realize that the line numbering used by the reviewers will always pertain to your original ACPD version of your manuscript, which will not be the same as that of either your revised version (where you show where you modified your manuscript), nor the final version. As your editor I spent a long time trying to match the line numbers of the three versions. In fact, on more than one occasion I found the line numbers you gave in your replies to the referees impossible to match. I do not have a good suggestion for this, but at least I would have appreciated if you had indicated to which version the line numbers in your replies referred (I presumed the final clean version but that did not appear to be always the case).

As for editorial comments, both referees have given you rather substantial lists of corrections, and I have several more. The sheer number of corrections indicates poor proof reading. I strongly recommend to have the final manuscript proofread once more by one of your co-authors.

Here is my list (where the line numbers refer to the clean manuscript version 4 of December 19, 2017.

L41: "causing" -> this having

l44: "and can be transported -> and **it** can be transported

l48: this is the first time for defining the term "ECCC" but you have to say so: "ECCC (Environment…)"

l68: "AOSR facilities are causing critical levels of NH3". I think I know what you mean but this formulation is wrong. Make it something like "…are responsible for NH3 reaching critical levels.."

l94: it would be interesting to know how much improvement there was by using the updated version of GEM-MACH in comparison to its original version. Was there a big improvement already, and hence is use of the bidi and dire versions maybe just a slight further improvement? Some numbers would be usefujl to remove any doubts here.

L110: "physics package, this component…" -> "physics package. This component…" (too long and convoluted sentence, cut in two)

L141: reference to "this issuer" is not possible, see (2) above.

L149-156: delete (repeat)

L2015: "which are based on.." -> ":which **is** basewd on.."

L221: Since our simulation occurred.." -> "Since our simulation pertained to.."

L280: don't you need also chemical data as input for the 12 day spin up? Need to say something about that.

L386: "but not shown in the time series" -> "but **are** not shown in the time series"

L387: "then NH3 concentrations seen in…" a little odd, because you just said in the previous line that they are not shown. Try another wording.

L398: this seems comparing apples and oranges: the bias in the new model vs ground data was comparable to the bias of the old model vs satellite data? Correct what you mean or delete this comparison.

L452: "The model also has" -> "The model output also has"

L506: why reverse in time? Make it "(12 Aug, 1 Sept, and 3 Sept, 2013)" (and change the order in the following text accordingly)

L538: "should improve"? You don't know this, it might go the other way! Change it "might improve"

L551-569: this part is mangled up and must be reformulated.

L648: why are you so sure that the Hsu and Clair (2015) number is correct, and your number is an underestimate? This needs a justification, even if you next give arguments why you number might be low. While at this, I note you have a reference to Hsu and Clair (2015) (l645) and Hsu and Clair (2016) (l648). On scanning the list of references I see that these references appear to be to two different papers, but the titles of these references are identical. Please review and correct!

L682: delete "greatly" (subjective and not defensible)

Page 31: mixture of the last page of references with Figure 1 and Table 1

Fig 6a-d and Fig 8c-d: "Rfire" (text embedded in the figure) should be "Rfire+bidi"

Table 2: l3 of the title: "|t|" ? I presume "t"?

---

## Author Response (AR2)

Point by point response to editor's minor revisions

Thank you for these additional corrections to the manuscript. Below, editor's comments are in itallic, and author responses are in normal font, starting with [CW].

*(1) You have used units of ppbv for NH3. I assume you know that ppbv is not a concentration unit (it is dimensionless). It is also not the proper SI unit (should be nmol/mole). I will not bother you with ordering to change "ppbv", but whenever in your text you use "ppbv", you should change the term "concentration" into "mixing ratio", or the more correct term "mole fraction"*

[CW] We replaced "concentration" by "volume mixing ratio" where appropriate, throughout the text and figure and table captions.

*(2) As for referencing other papers that (will) belong to the AOSR special issue you should use the actual full ACP reference rather than "this issue". ACP papers are published on line when accepted, and will not be hold back to appear all at once with consecutive numbering, so the term "this issue" does not work. Furthermore, I noticed somewhere a reference to a manuscript in the ACPD phase. That is somewhat acceptable (you refer in that case not to a peer reviewed publication, although the manuscript has been accepted for review, which is something entirely different!) but here even more reference to "this issue of ACDP" is meaningless (manuscripts in ACPD will not appear in a hardcopy version, only online.*

[CW] We removed any text mentioning "this issue", and have simply stated the ACP or ACPD reference.

*(3) I applaud you for the consecutively line numbering of your manuscript. However, you have to realize that the line numbering used by the reviewers will always pertain to your original ACPD version of your manuscript, which will not be the same as that of either your revised version (where you show where you modified your manuscript), nor the final version. As your editor I spent a long time trying to match the line numbers of the three versions. In fact, on more than one occasion I found the line numbers you gave in your replies to the referees impossible to match. I do not have a good suggestion for this, but at least I would have appreciated if you had indicated to which version the line numbers in your replies referred (I presumed the final clean version but that did not appear to be always the case).*

[CW] Indeed, I was confused too, as one referee quoted line numbers from my original submitted manuscript, and the other referee quoted line numbers from the manuscript I submitted with the technical corrections. There was only one comment that I couldn't find the referenced text in either of those. Anyway, when replying, I was referring to the new submitted manuscript for line numbers (unless otherwise stated "in original manuscript" – meaning the one the referee was referring to), but may have made some mistakes along the way.

*As for editorial comments, both referees have given you rather substantial lists of corrections, and I have several more. The sheer number of corrections indicates poor proof reading. I strongly recommend to have the final manuscript proofread once more by one of your co-authors.*

[CW] I have done a thorough proof read of this latest submission, and so has my co-author Paul Makar.

*Here is my list (where the line numbers refer to the clean manuscript version 4 of December 19, 2017.*
*L41: "causing" -> this having*

[CW] Done.

*l44: "and can be transported -> and **it** can be transported*

[CW] Done.

*l48: this is the first time for defining the term "ECCC" but you have to say so: "ECCC (Environment…)"*

[CW] Both here and on line 32 (of the Dec 19[th] manuscript), "Environment and Climate Change Canada" is the author of the references. The first use of the department NOT as a reference is on line 105 (of Dec 19[th] manuscript), so that is why we have defined the acronym there instead. It doesn't seem right to define an acronym in the middle of a reference. Nothing changed.

*l68: "AOSR facilities are causing critical levels of NH3". I think I know what you mean but this formulation is wrong. Make it something like "…are responsible for NH3 reaching critical levels.."*

[CW] Rephrased to "It is important to understand if the AOSR facilities are causing NH$_3$ to reach critical levels, and if not, …"

*I94: it would be interesting to know how much improvement there was by using the updated version of GEM-MACH in comparison to its original version. Was there a big improvement already, and hence is use of the bidi and dire versions maybe just a slight further improvement? Some numbers would be usefujl to remove any doubts here.*

[CW] The updated version of GEM-MACH is our "base" scenario (=GEM-MACHv2). While we didn't compare our base scenario to the same TES observations to directly see the improvement over the previous Shephard et al (2015) study, we do know that TES and CrIS have very similar measurements of NH3 in that region, which we stated on line 484-486 (Dec 19[th] version) and from CrIS validation work by Mark Shephard (some of which was shown in Shephard and Cady-Peirera (2015), Fig 10, top panel). And we see that our base scenario has a very similar bias to CrIS, as the old model (GEM-MACHv1.5.1) had with TES, and the text has been updated to say this. Therefore, there is little-to-no improvement between the updated version of the model (base, GEM-MACHv2), and the original version in Shephard et al (2015, GEM-MACHv1.5.1).

*L110: "physics package, this component…" -> "physics package. This component…" (too long and convoluted sentence, cut in two)*

[CW] We have shortened the sentence by replace "the meteorological model" with GEM, which was defined in the previous sentence. It now reads as: "This means that the chemical processes of the model (…) are imbedded within GEM's physics package, which in turn is imbedded within GEM's dynamics package, the latter handling chemical tracer advection."

*L141: reference to "this issuer" is not possible, see (2) above.*

[CW] Removed all instances of "this issue".

*L149-156: delete (repeat)*

[CW] Deleted.

*L205: "which are based on.." -> ":which **is** basewd on.."*

[CW] Done.

*L221: Since our simulation occurred.." -> "Since our simulation pertained to.."*

[CW] Done.

*L280: don't you need also chemical data as input for the 12 day spin up? Need to say something about that.*

[CW] lines 290-304 (of the Dec 19[th] manuscript) explain how the chemical data were treated, however, we have reorganized this section in order to be more clear. Please see the revised manuscript, Section 2.4.

*L386: "but not shown in the time series" -> "but **are** not shown in the time series"*

[CW] Done.

*L387: "then NH3 concentrations seen in…" a little odd, because you just said in the previous line that they are not shown. Try another wording.*

[CW] Removed the word "seen" so that it's less confusing.

*L398: this seems comparing apples and oranges: the bias in the new model vs ground data was comparable to the bias of the old model vs satellite data? Correct what you mean or delete this comparison.*

[CW] You've understood correctly, and we removed this text here when discussing the ground data. Some new text (related to the Shephard et al, 2015 study) was added in Section 4.3, when discussing the satellite results, which makes more sense.

*L452: "The model also has" -> "The model output also has"*

[CW] Done.

*L506: why reverse in time? Make it "(12 Aug, 1 Sept, and 3 Sept, 2013)" (and change the order in the following text accordingly)*

[CW] We re-ordered the dates here, however, the figures and discussion remain in the reverse order in order to more clearly facilitate the discussion (first discussing northern bidi performance, then southern bidi performance, and then the model's fire performance).

*L538: "should improve"? You don't know this, it might go the other way! Change it "might improve"*

[CW] Changed to "is likely to" because of the evidence discussed in this section.

*L551-569: this part is mangled up and must be reformulated.*

[CW] Done – see Section 4.3.2; the arguments have been made more concise and are no longer in point form, refering back to the different explanations.

*L648: why are you so sure that the Hsu and Clair (2015) number is correct, and your number is an underestimate? This needs a justification, even if you next give arguments why you number might be low. While at this, I note you have a reference to Hsu and Clair (2015) (l645) and Hsu and Clair (2016) (l648). On scanning the list of references I see that these references appear to be to two different papers, but the titles of these references are identical. Please review and correct!*

[CW] Removed "our underestimate", and rephrased so that it doesn't seem like theirs is right and ours is wrong. The two Hsu references have been corrected.

*L682: delete "greatly" (subjective and not defensible)*

[CW] Deleted.

*Page 31: mixture of the last page of references with Figure 1 and Table 1*

[CW] This is an odd thing that latex does, but will not be an issue in the final paper whereby the latex and figure documents are submitted separately.

*Fig 6a-d and Fig 8c-d: "Rfire" (text embedded in the figure) should be "Rfire+bidi"*

[CW] Done.

*Table 2: l3 of the title: "|t|" ? I presume "t"?*

[CW] Caption unchanged. We mean that if the magnitude of $t$ is < 1, then … . Therefore, we put the absolute value sign around $t$ because it could be positive or negative, but it's the magnitude that counts.

[revised manuscript text omitted]

---

## Author Response (AR3)

Point by point response to editor's corrections

*Oh, those "concentrations"! Line 646 in the final version, delete "concentrations" after NH4+ (double up).*
*Also Figure 4: delete the label for the y-axis (4a is in VMR, 4b-d in concentration); you give the labels in the text below the figure...*
*Other than that, accepted and on to publication! Truly a nice story, I liked it.*
*Jan Bottenheim*

[CW] Thank you for catching those, and for your work as co-editor! I have removed the y-label of Fig 4, and the second mention of the word "concentrations" in the text.